# Formaldehyde total column densities over Mexico City: comparison between MAX-DOAS and solar absorption FTIR measurements

Claudia Rivera Cárdenas[1], Cesar Guarín[1,2], Wolfgang Stremme[1], Martina M. Friedrich[1,3], Alejandro Bezanilla[1], Diana Rivera Ramos[1], Cristina A. Mendoza-Rodríguez[1], Michel Grutter[1], Thomas Blumenstock[4], and Frank Hase[4]

[1]Centro de Ciencias de la Atmósfera, Universidad Nacional Autónoma de México, Mexico City, Mexico
[2]Departamento de Física, Universidad Autónoma Metropolitana, Mexico City, Mexico
[3]Belgian Institute for Space Aeronomie (BIRA-IASB), Brussels, Belgium
[4]Institute of Meteorology and Climate Research, Karlsruhe Institute of Technology, Karlsruhe, Germany

**Correspondence:** claudia.rivera@atmosfera.unam.mx

**Abstract.** Formaldehyde (HCHO) total column densities over the Mexico City Metropolitan Area (MCMA) were retrieved using two independent measurement techniques: Multi Axis - Differential Optical Absorption Spectroscopy (MAX-DOAS) and Fourier Transform Infrared (FTIR) Spectroscopy. For the MAX-DOAS measurements, the software QDOAS was used to calculate differential Slant Column Densities (dSCDs) from the measured spectra and subsequently the Mexican MAX-DOAS Fit retrieval code (MMF) to convert from dSCDs to Vertical Column Densities (VCDs). The direct-solar absorption spectra measured with FTIR were analyzed using the PROFFIT retrieval code. Typically the MAX-DOAS instrument reports higher VCDs than those measured with FTIR, in part due to differences found in the ground-level sensitivities as revealed from the retrieval diagnostics from both instruments, as the FTIR and the MAX-DOAS information do not refer exactly to the same altitudes of the atmosphere. Three MAX-DOAS datasets using measurements conducted towards the east, west or both sides of the measurement plane were evaluated with respect to the FTIR results. The retrieved MAX-DOAS HCHO VCDs where 5%, 9% and 28% larger than the FTIR which, supported with satellite data, indicates a large horizontal inhomogeneity in the HCHO abundances. The temporal change in the vertical distribution of this pollutant, guided by the evolution of the mixing layer height, affects the comparison of the two retrievals with different sensitivities (total column averaging kernels). In addition to the reported seasonal and diurnal variability of HCHO columns within the urban site, background data from measurements at a high-altitude station, located only 60 km away are presented.

## 1 Introduction

Megacities are in constant evolution, exhibiting continuous changes in territorial extension, population size and spatial redistribution, as well as in the types of socio-economical activities performed every day. In many cases the spatial growth is uneven, resulting on areas of the city more prone to emissions or accumulation of pollutants due to chemical transformations or transport patterns influenced by meteorological conditions. For the specific case of the Mexico City Metropolitan Area (MCMA), the urban sprawl observed over the years has been topographically influenced causing redensification processes due

to the space-limited valley location of the MCMA (Taubenböck et al., 2012). The associated natural emissions in and around the MCMA (Central Mexican matorral and Trans-Mexican Volcanic Belt pine-oak forests), along with the daily activities conducted by the MCMA population, nearly 22 million inhabitants, base year 2018, according to the Demographic Statistics Database of the United Nations Statistics Division (UN, 2019) are considered as the main drivers of the enhancement of HCHO in the atmospheric column over the MCMA.

Formaldehyde (HCHO), a hazardous pollutant present mostly in the lower troposphere, is the most abundant carbonyl compound found in urban areas such as Mexico City. Due to its short lifetime of only a few hours, the quantitative determination of this gas gives an idea of the distribution of its sources (Stavrakou et al., 2009). It can be directly emitted by several sources including automobile exhaust, natural gas combustion, biomass burning, building materials, personal care and cleaning products, among many others. It can likewise be emitted directly by vegetation, although in low concentrations (Kesselmeier et al., 1997). Formaldehyde is mainly formed in the air from the oxidation and degradation of methane and many non-methane volatile organic compounds (NMVOCs), both of biogenic and anthropogenic origin (Finlayson-Pitts and Pitts, 2000; Solal et al., 2008), and is a radical source involved in urban tropospheric chemistry and ozone formation (Lei et al., 2009).

There are two main reactions that HCHO undergoes in the atmosphere, photolysis and reaction with OH (Seinfeld and Pandis, 2012), leading to the formation of carbon monoxide which in turn produces tropospheric ozone.

$$\mathrm{HCHO} + h\nu \quad \rightarrow \quad \mathrm{H}\cdot + \mathrm{HCO} \tag{R1}$$
$$\rightarrow \quad \mathrm{H_2} + \mathrm{CO} \tag{R2}$$
$$\mathrm{HCHO} + \mathrm{OH}\cdot \quad \rightarrow \quad \mathrm{HCO} + \mathrm{H_2O} \tag{R3}$$

Deriving the global burden and emissions of many NMVOCs is a real challenge from the limited observations available, however, satellite HCHO observations can constrain their emissions in global chemistry transport models and thus provide a better understanding of their spatial distributions and temporal variability.

Remote sensing techniques are a useful complement in the quantification of gases by measuring the total atmospheric column amount along a line-of-sight. Spectrometers installed on board satellites, aircraft, balloons, vehicles or ground-based stations have the capacity to determine the atmospheric composition of gases and particles by observing their characteristic interaction with the radiation field. A common technique deployed from the ground is solar absorption Fourier Transform Infrared (FTIR) spectroscopy, capable of quantifying vertical column densities and profiles of a wide range of gases (Hase et al., 2004; Stremme et al., 2009; Bezanilla et al., 2014). A comprehensive study by Vigouroux et al. (2018) was carried out for HCHO to harmonize the retrieval settings of 20 ground-based FTIR instruments contributing to the Network for the Detection of Atmospheric Composition Change (NDACC). HCHO can also be measured with the Differential Optical Absorption Spectroscopy (DOAS) technique (Platt and Stutz, 2008). The retrieved HCHO slant columns measured with 9 DOAS instruments in the Multi-AXis configurations (MAX-DOAS) were inter-compared in the CINDI (Cabauw Intercomparison campaign of Nitrogen Dioxide measuring Instruments) field campaign and presented consistent results (Pinardi et al., 2013). During the CINDI-2 campaign, considerable differences on the Differential Slant Column Densities retrieved from measurements conducted by different in-

struments were identified (Kreher et al., 2020). The advantage of the MAX-DOAS technique in comparison to the zenith-sky DOAS approach is that vertical column densities can be retrieved with some information on the vertical distribution in the lower troposphere (Platt and Stutz, 2008).

Some comparisons between FTIR and MAX-DOAS instruments have been done in the past. Surface HCHO was measured with two spectroscopic techniques by Grutter et al. (2005) in the Mexico City downtown area during 2003, reporting monthly averages that ranged between 12.7 and 23.9 ppm. In that study, the open-path FTIR and Long Path-DOAS derived products agreed within 15%. Vigouroux et al. (2009) found a good agreement in HCHO columns retrieved from solar absorption FTIR and MAX-DOAS measurements during campaigns performed in 2004 and 2007 on Reunion Island. The ground-based observations were also compared with the SCIAMACHY satellite product and from results of a chemical transport model. Franco et al. (2015) retrieved vertical profiles from MAX-DOAS and FTIR at the Jungfraujoch station, which were subsequently compared to two chemical transport models (GEOS-Chem and IMAGES v2), concluding that both measurement techniques (FTIR and MAX-DOAS) can be considered as providing complementary information for the retrieval of HCHO above the Jungfraujoch station. Tirpitz et al. (2020) found very good agreement of MAX-DOAS retrieved VCDs compared to direct sun DOAS (an average root-mean-square difference of $1.4 \times 10^{15}$ molec/cm$^2$) as found during CINDI2. Another study by Garcia et al. (2006) aimed at evaluating the relative primary (directly emitted) to secondary (photochemically produced) contributions to ambient HCHO concentrations in Mexico City. By using a statistical analysis and carbon monoxide and glyoxal as gas-phase tracers of primary and secondary HCHO respectively, they found that during daytime the photochemically produced HCHO may account for up to 80% while during the night and before sunrise the primary sources, such as vehicle emissions, dominate the HCHO concentrations at the surface. In a study conducted by Lei et al. (2009), the impact of primary HCHO on pollution in Mexico City was analyzed. The authors indicate that HCHO emitted by primary sources dominates the concentration of this carbonyl both in the morning and at night, and HCHO decreases by approximately 1/3 in the afternoon.

In this study, we use a time series of more than 7 years to perform an unprecedented comparison (in terms of length and location) of the HCHO total vertical column amount measured with two independent techniques. Retrieval diagnostics from both the MAX-DOAS and FTIR results are used to characterize the difference in both measurement techniques and to improve the agreement and correlation between coincident data pairs (section 3.3). The seasonal and diurnal variability of HCHO columns is reported from a measurement site in the Mexico City urban area, as well as from a remote site in a high-altitude station located only 60 km away (section 3.1). Together with space-based observations (section 3.2), these results do not only serve to understand the local conditions in which this pollutant is emitted, produced and transported within the Mexico City Metropolitan Area (MCMA), but will also provide confidence in the validation activities of model results as well as of the current and future satellite missions.

## 2  Methodology

In this section we describe the two independent measurement techniques, based on FTIR spectroscopy and MAX-DOAS, used to retrieve HCHO vertical column densities over two measurement sites. One of the sites is in the south of the MCMA,

at the Universidad Nacional Autónoma de México (UNAM) Campus -National Autonomous University of Mexico- on the roof of the Centro de Ciencias de la Atmósfera (CCA-UNAM, 19.32 Lat, -99.17 Lon, 2280 m a.s.l.) -Center for Atmospheric Sciences-. The other site is the Altzomoni Atmospheric Observatory, a high-altitude research facility located in the Iztaccíhuatl-Popocatépetl National Park, 60 km south-east of Mexico City (19.12 Lat, -98.66 Lon, 3985 m a.s.l.). Both the UNAM and

Altzomoni stations are part of the Red Universitaria de Observatorios Atmosféricos (RUOA) national monitoring network -University Network of Atmospheric Observatories- and the Altzomoni station is also part of the NDACC international network.

## 2.1   Solar absorption FTIR measurements

The UNAM station is equipped with a Fourier transform infrared spectrometer (FTIR) from Bruker Optics (model Vertex 80) that measures solar absorption spectra at different spectral regions with Mercury Cadmium Telluride (MCT) and Indium

Gallium Arsenide (InGaAs) detectors and 5 band-pass filters placed on a rotating wheel. The interferometer has a maximal optical-path difference of 12 cm and continually records spectra at $0.075$ cm$^{-1}$ resolution. The light from the Sun is followed with a home-built solar tracker mounted inside an astronomical dome that is automated so that it can measure from sunrise to nightfall, as long as there are no clouds and ensures measurements are conducted with a 7 milliradians field of view (smaller than the solar disk). The solar tracker has been continuously improved over the years and uses, in its current version, a camera

mounted behind a beam-splitter in the optical bench that is used as feedback to optimize the pointing to the Sun (Bezanilla et al., 2014).

    At the Altzomoni remote site, a high-resolution FTIR (Bruker Optics, IFS120/5) is operated remotely with a microwave antenna that allows us to have communication with the station. This instrument allows a maximal optical-path difference of 257 cm, recording spectra typically at $0.005$ cm$^{-1}$ resolution, and its solar tracker uses an astronomical telescope-mount that is

controlled by the Camtracker software (Gisi et al., 2011; Gisi, 2012). Further details about the instrumental setup are provided elsewhere (Baylon et al., 2017; Plaza-Medina et al., 2017; Taquet et al., 2019).

    Vertical Column Densities (VCD) are retrieved from FTIR solar absorption spectra in 4 spectral microwindows in the region between 2763-2782 cm$^{-1}$, using the spectroscopic line-data compilation *AMT16* (Toon et al., 2016), available at http://mark4sun.jpl.nasa.gov/toon/linelist/linelist.html (last access: 20 January 2020), a Tikhonov L1 constraint and an *a pri-*

*ori* VMR profile taken as a 41-year mean (1980-2020) from the Whole Atmosphere Community Climate Model (WACCAM, version 4) model run over the particular measurement sites. Although this *a priori* profile would correspond to the background HCHO concentration and not to the polluted conditions in Mexico City, this will not affect the resulting profile as a Thikonov-L1 constraint is used and just the relative shape of the vertical *a priori* profile is relevant for the constraint. The retrieval strategy and error analysis follows the recipe described by Vigouroux et al. (2018). The data are processed using the retrieval

code PROFFIT9 (Hase et al., 2004) and the random errors at UNAM and Altzomoni are estimated to be around $5\%$ ($1.1 \times 10^{15}$ molec/cm$^2$) and $10\%$ ($0.2 \times 10^{15}$ molec/cm$^2$) of the corresponding mean columns ($22.1 \times 10^{15}$ molec/cm$^2$, $2.18 \times 10^{15}$ molec/cm$^2$) (Vigouroux et al., 2018).

## 2.2 The MAX-DOAS measurements

A MAX-DOAS instrument, designed and built by the Spectroscopy and Remote Sensing Group at CCA-UNAM, was used to conduct sky measurements in the UV-Vis region of the electromagnetic spectrum. The MAX-DOAS instrument, which is collecting data since 2013, is installed on the roof of CCA-UNAM (same location as the FTIR-Vertex instrument) and forms part of a small network (Arellano et al., 2016) of MAX-DOAS instruments that covers part of the MCMA.

The MAX-DOAS has a theoretical field of view of 0.31° (Arellano et al., 2016) and performs measurements sequences with a telescope's azimuth angle of 85°with respect to the north. Each elevation scan of the MAX-DOAS measurements starts with a zenith measurement. This is followed by a number of measurements towards different elevation angles, starting at low angles, all towards the same azimuthal direction. Approaching the zenith direction, the scan is continued in reverse order for the elevation angles towards the opposite azimuthal direction, from large elevation angles towards small elevation angles. The measurement sequence, as specified in Friedrich et al. (2019) is: 90°zenith, 0, 2, 6, 13, 23, 36, 50, 65, 82°W, 82, 65, 50, 36, 23, 13, 6, 2, 0°E. At the end of each scan, a dark spectrum is taken (closed shutter). With this setup, a complete scan takes about 7 min. The dark spectrum is subtracted from all other spectra measured during the sequence (including the zenith spectrum). A detailed instrument description and measurement strategy can be found in Arellano et al. (2016) and Friedrich et al. (2019).

Before conducting retrievals, spectra are filtered with the objective to remove all spectra either with too low light conditions (10% or less of the maximum possible intensity level) or saturated spectra in the retrieval region. Differential Slant Column Densities (dSCDs) are retrieved from the collected spectra at different elevation angles, following a DOAS approach using the QDOAS software developed at the Belgian Institute for Space Aeronomie (Danckaert et al., 2013). A wavelength calibration was conducted in QDOAS by applying a nonlinear least-squares fit to a solar atlas (Chance and Kurucz, 2010). HCHO was retrieved in the 324.6 to 359 nm wavelength range, a polynomial order 5 was used along with an offset order 1 (linear offset) (Hendrick et al., 2016; Pinardi, 2017). Specific details about the used cross-sections are provided in Table 1. Cross-sections were convolved with the instrumental slit function and a wavelength calibration file created using a mercury lamp. $O_4$ was retrieved in the 336 to 390 nm wavelength range, following settings described in Friedrich et al. (2019).

**Table 1.** DOAS analysis settings for HCHO slant column density retrieval.

| Parameter | Specification |
| --- | --- |
| Fitting interval | 324.6 to 359 nm |
| Cross sections | |
| HCHO | Meller and Moortgat (2000) at 298K |
| $O_3$ | Serdyuchenko et al. (2014) at 223 and 243K |
| $NO_2$ | Vandaele et al. (1998) at 298K |
| BrO | Fleischmann et al. (2004) at 223K |
| $O_4$ | Thalman and Volkamer (2013) |
| Ring spectrum | Calculated with QDOAS according to Chance and Spurr (1997) and normalized as in Wagner et al. (2009) |

HCHO VCDs were retrieved using the MMF code (Friedrich et al., 2019). MMF uses HCHO dSCDs and converts them into VCDs in a two steps process for each scan: first, the $O_4$ slant column density information is used to retrieve an aerosol profile. In the second step, the retrieved aerosol profile information is used together with the HCHO dSCDs to retrieve the trace gas profile. Both parts follow a procedure that consists of a forward model and an inversion algorithm. A constrained least squares fit is used in both steps, but the aerosol retrieval uses Tikhonov regularization and the trace gas retrieval uses optimal estimation. The forward model uses the radiative transfer code VLIDORT v2.7 (Spurr et al., 2001; Spurr, 2006, 2013). The inputs to VLIDORT are calculated using temperature and pressure information from daily radiosonde measurements and aerosol single scattering optical depths and asymmetry factors from the AERONET (Aerosol Robotic Network) data base for Mexico City.

To run MMF retrievals, the absorption cross section was taken at a wavelength in between the range of the wavelength interval used for the QDOAS retrieval: for $O_4$ retrieval it was 360 nm and for HCHO it was 338 nm. For aerosol, the *a priori* profile shape was taken from 1 year of ceilometer data, averaged each hour. The *a priori* aerosol data for total optical depth were time-interpolated from the co-located AERONET station in Mexico City. For HCHO retrieval, a single *a priori* profile and covariance matrix taken from the chemical transport model WRF-Chem was used. The surface albedo used in this study was set to 0.07. For the retrieved HCHO MAX-DOAS VCDs, the calculated noise error of the mean column is 5.8% while the systematic error due to uncertainty in the spectroscopy is 2.2%.

Three different versions of HCHO VCDs were retrieved using the MMF code: **V1** retrieved VCDs from MAX-DOAS measurements conducted towards the east (telescope's azimuth angle of 85°with respect to the north), **V2** retrieved VCDs from MAX-DOAS measurements conducted towards the west (telescope's azimuth angle of 265°with respect to the north) and **V3** retrieved VCDs from MAX-DOAS measurements conducted towards both sides of the scanning plane. To simplify terminology, for the remainder of the manuscript version **V1** will be referred as "east", version **V2** will be referred as "west" and version **V3** will be referred as "both".

For **V1**, **V2** and **V3** the same *a priori* is used both for the trace gas and for the aerosol. For **V3**, the "scan" is simply treated as consisting of two different azimuth directions. The **V1**, **V2** and **V3** retrievals are performed independent of each other and differ in the definition of a "scan", where **V3** contains all pointing directions from **V1** and **V2** together. A single vertical profile is retrieved in both directions for **V3**, so assuming horizontal homogeneity. This assumption clearly is not fulfilled, however, it is also not fulfilled in a single viewing direction since the effective light path is around 5-20 km. As pointed out in the manuscript, the advantage of using both directions is a higher information content, while the disadvantage is a more rigorous break down of the homogeneity assumption.

## 3  Results

### 3.1  Diurnal and seasonal variability of HCHO in Mexico City

A large data set of measurements taken at the UNAM site between January 2013 and May 2020 allowed us to study the diurnal and seasonal variability of HCHO. Figure 1 shows the time series of HCHO VCDs hourly means from the MAX-DOAS

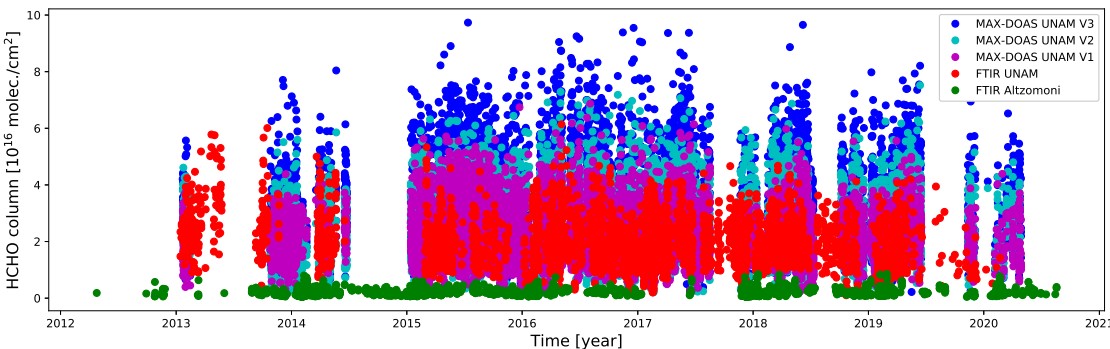

**Figure 1.** Hourly means of HCHO vertical column densities (VCDs) from the MAX-DOAS instrument at UNAM **(V1)** (magenta), **(V2)** (cyan), **(V3)** (blue), solar absorption FTIR instrument at UNAM (red) and at Altzomoni (green). The measurements in Mexico City at the UNAM-campus (red, magenta, cyan & blue) are one order of magnitude higher than the background measurements at the mountain site Altzomoni (green). The same applies to their variance. The difference between the MAX-DOAS and FTIR measurements at UNAM is mainly explained by their averaging kernel and is discussed in detail in the text. The altitude ranges covered by each instrument are FTIR UNAM VCDs 2-16 km, FTIR Altzomoni VCDs 4-16 km and MAX-DOAS UNAM VCDs 2-5 km.

(**V1, V2 and V3**) and from FTIR instruments located at this urban site. Additionaly, HCHO VCDs from a high resolution FTIR instrument were measured from the remote site at Altzomoni providing relevant information about the variability of the background concentrations (see section 3.4). One can see that both instruments located at UNAM report values in the same

order of magnitude, however, higher values in MAX-DOAS measurements than the FTIR instrument are apparent. This is in part attributed to the larger ground level sensitivity of the MAX-DOAS instrument with respect to the FTIR as will be shown later. Compared to the instrument at UNAM, the FTIR instrument located at Altzomoni, presents lower HCHO VCDs hourly means values. This is to be expected since that instrument is located at a higher altitude level (>1700 m higher than UNAM) and therefore above the mixed-layer most of the time and thus probes cleaner atmospheric columns, as long as there is no

upslope transport.

Figure 2 shows the diurnal cycle of hourly averaged HCHO from FTIR (red), MAX-DOAS (**V3**) (blue), MAX-DOAS **V1** (magenta) and MAX-DOAS **V2** (cyan) measurements at UNAM. The VCD hourly averages from the MAX-DOAS instrument are in general larger than those from the FTIR instrument (0 to 38% for **V1**, 15 to 47% for **V2** and 29 to 61% for **V3**), however, both exhibit a similar pattern throughout the day. The results from the FTIR measurements show a steady increase

of HCHO VCDs from early morning until 13 h LT, time when the HCHO VCDs start to slightly decrease until the end of the measurement day. The results from the MAX-DOAS instrument also show a decrease after 13 h LT, but show an increase at 16 h LT, likely due to traffic during rush hour. A decrease of primary HCHO (to 32% or less) in the afternoon was reported by Lei et al. (2009), a behavior not necesarily observed in the measurements reported in this study for the MAX-DOAS datasets. The slight increase of the HCHO column from 16 h LT (MAX-DOAS datasets) could be an indication of the contribution of secondary HCHO becoming more important in the atmospheric column in the afternoon. It should however be noted that the

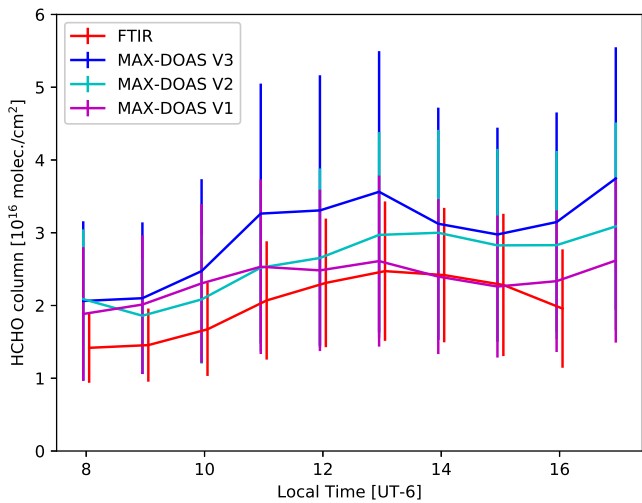

**Figure 2.** Diurnal cycle from hourly averaged HCHO VCDs from FTIR (red), MAX-DOAS **V3** (blue), MAX-DOAS **V1** (magenta) and MAX-DOAS **V2** (cyan) measurements at UNAM. Vertical lines represent the standard deviation of all conducted measurements during that hour.

conclusions presented by Lei et al. (2009) are based on a modeled three-day episode case study constrained by ground-based measurements (conducted during the MCMA-2003 field campaign) at one site, while the results presented in the current study are based on more than 8 years of measurements of the amount of HCHO in the tropospheric colum. This situation could 5   explain the differences on the observations reported in this study and by Lei et al. (2009).

In Figure 3, the seasonal (monthly average) HCHO VCDs cycle of FTIR (red), MAX-DOAS (**V3**) (blue), MAX-DOAS **V1** (magenta) and MAX-DOAS **V2** (cyan) measurements at UNAM is presented. As in the case of the diurnal cycle, MAX-DOAS HCHO VCDs are larger than the ones reported by the FTIR measurement technique (2 to 35% for **V1**, 17 to 51% for **V2** and 23 to 75% for **V3**), nevertheless, the two datasets are within each other temporal variability. Both instruments show two maxima: 10   May and September for the MAX-DOAS and May and October for the FTIR. In addition, both instruments present the lowest HCHO VCDs values during January and December.

### 3.2   HCHO horizontal distribution from OMI satellite observations

In order to assess the horizontal inhomogeneity of HCHO in the MCMA, an average distribution map of HCHO was constructed from data between 2005 and 2018 from the Ozone Monitoring Instrument (OMI) satellite instrument and is presented in Figure 15   4. The OMHCHO Version-3 data product (Chance, 2007) was downloaded from the EARTHDATA web portal. The map was generated using the *ReferenceSectorCorrectedVerticalColumn* field from the SAO OMI product (González Abad et al., 2015). Only data with cloud fraction of 20% or less and the field *MainDataQualityFlag* set to 0 was used. From this average

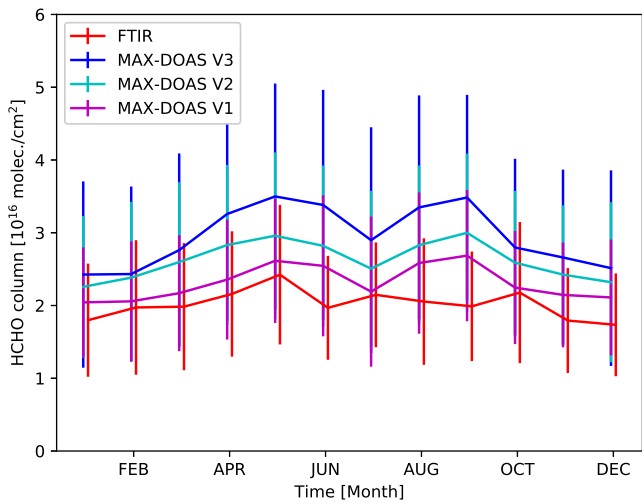

**Figure 3.** Seasonal FTIR (red), MAX-DOAS **V3** (blue), MAX-DOAS **V1** (magenta) and MAX-DOAS **V2** (cyan) cycles at UNAM. Vertical lines represent the standard deviation of all conducted measurements during each month.

HCHO distribution map, a strong horizontal inhomogeneity over the MCMA is evident and motivates the investigation of the differences in computing VCDs using MAX-DOAS measurements conducted towards the eastern (**V1**) or western (**V2**) sides of the scanning plane, or using all available measurements (**V3**), as will be described below.

## 3.3 MAX-DOAS vs FTIR comparison at UNAM

### 3.3.1 Differences in the MAX-DOAS viewing directions

A detailed comparison between VCDs retrieved using the MAX-DOAS and FTIR measurement techniques was conducted and is explained in this section. The correlation between the coincident hourly mean vertical columns from FTIR and MAX-DOAS measured at UNAM are shown in Figure 5. The plots shown in the left column contain the retrieved VCDs without any correction. Four different data sets are presented in the different rows from top to bottom corresponding four product versions: **V1** retrieved VCDs from MAX-DOAS measurements conducted towards the east, **V2** retrieved VCDs from MAX-DOAS measurements conducted towards the west, **V3** retrieved VCDs from MAX-DOAS measurements conducted towards both sides of the scanning plane and in the fourth row **V1** retrieved VCDs from MAX-DOAS measurements conducted towards the east during the morning hours and **V2** retrieved VCDs from MAX-DOAS measurements conducted towards the west during the afternoon hours. The correlation coefficient is provided, along with the linear regression information when forced to zero (red) and not constrained (green). Black lines represent the 1:1 relation. The resulting total column averaging kernel from the retrievals of the FTIR and MAX-DOAS data sets are presented in the third column of Figure 5, which already explains partly the relation between the vertical columns. The total column averaging kernel is the sum of the rows of the averaging kernel

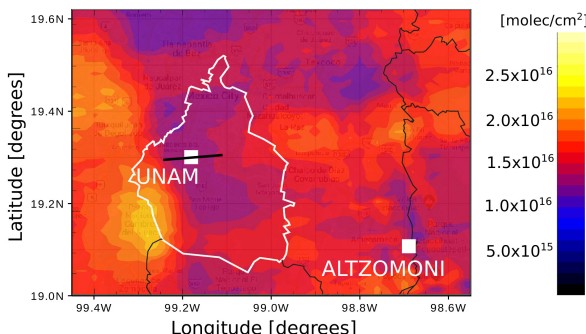

**Figure 4.** Average HCHO total column distribution map over the MCMA between 2005 and 2018. The columnar HCHO distribution is reconstructed from OMI measurements on board the Aura satellite with a daily Mexico City overpass at around 14:00 LT (spatial distribution is only representative for this time). Color bar units are in molecules/cm$^2$. The UNAM and Altzomoni measurement sites are marked with a square, the region comprising Mexico City is indicated in white, and the measurement directions are indicated with a black line over the UNAM site.

in partial columns [molecules/cm$^2$ / molecules/cm$^2$]. The averaging kernel of the FTIR is lower than 1.0 within the mixed layer (2280-4000 m a.s.l.), containing the largest amount of HCHO, and thus the VCD is underestimated. It can be seen that the **V1** data set has a better agreement (slope closer to 1) than **V2** and **V3**. It is interesting to note that particularly the VCDs

retrieved from MAX-DOAS using both measurement directions (third row) have an enhanced sensitivity in comparison to the FTIR retrieval, which partly explains the slope 1.28 and the 28% overestimation with respect to the FTIR retrievals. The comparison between FTIR and MAX-DOAS based on the coincidence of the measurement geometry (**V1** retrieved VCDs from MAX-DOAS measurements conducted towards the east during the morning hours and **V2** retrieved VCDs from MAX-DOAS measurements conducted towards the west during the afternoon hours) presented in the fourth row provides the best agreement

among the datasets.

For the correlation plots presented in the middle column of Figure 5, the retrieved MAX-DOAS profiles were smoothed with the FTIR averaging kernels resulting typically in lower MAX-DOAS VCD values. This smoothing process simulates how the HCHO profile retrieved by the MAX-DOAS should be seen by the FTIR instrument. The smoothed profiles are calculated by multiplying the averaging kernel of the FTIR retrieval with the retrieved MAX-DOAS profile (Rodgers, 2000). For the **V3**

data set, the smoothing by the FTIR kernel improves the slope from 1.28 to 1.11, much more might not be expected as the vertical information in the MAX-DOAS profiles is limited to less than two degrees of freedom (average values being 0.692 for **V1**, 0.782 for **V2** and 0.970 for **V3**) and do not represent the true atmospheric profile, while the average FTIR degrees of freedom is 1.0 for the UNAM site and 1.1 for the Altzomoni site (Vigouroux et al., 2018). For a fair comparison, the effect of

the different *a prioris* (Figure 6) in the retrieval is taken into account and the new *a priori* for both retrievals is the average of the MAX-DOAS profile retrieval of **V3**.

To further investigate the large HCHO inhomogeneity already shown in Figure 4, VCDs from the MAX-DOAS products
using different viewing directions (**V1**, **V2** and **V3**) were analyzed independently. Figure 7 shows correlation plots between the coincident HCHO VCDs when using dSCDs measured towards the east (a) or the west (b) compared to HCHO VCDs computed using both sides of the scanning plane. Black lines represent the 1:1 relation and a linear regression not constrained to zero is shown in green. The correlation between data sets for the east (**V1**) and both sides (**V3**) result in a Pearson's correlation coefficient of 0.887 with a slope of 0.630 and an offset of $6.781 \times 10^{15}$ molec/cm$^2$. When comparing west **V2** versus **V3**, the
calculated Pearson's correlation coefficient is 0.908 with a slope of 0.722 and an offset of $5.616 \times 10^{15}$ molec/cm$^2$. VCDs retrieved using measurements from both sides of the scanning plane are in general larger than VCDs retrieved using data from measurements of only one of the sides. This result can be explained by the larger amount of information available for the retrievals when dSCDs in different elevation angles and both scanning directions are used (average values being 0.692 for **V1**, 0.782 for **V2** and 0.970 for **V3**) (Figure 8). However, a conclusive explanation from this analysis cannot be derived without
investigating the time-dependent differences observed using different viewing directions.

### 3.3.2   Time-dependent differences in the MAX-DOAS viewing directions

The hourly differences between VCDs computed using the eastern or western sides of the scanning plane is investigated further, therefore simulated VCDs were calculated in order to compare them with measured VCDs. Simulated VCDs east-west differences are the result of the different amount of information in the retrievals in **V1** and **V2**. The true profile has much
higher HCHO concentrations in the polluted mixing layer than what the *a priori* reflects. The retrieval using both sides of the measurement plane contains more information originating from the measurements and allows the result on an optimal-estimation based retrieval to be less close to the *a priori*. The **V1** and **V2** retrievals do not always have the same amount of information, as the filtering criteria for the spectra do not act similarly throughout the day and spectral measurements are selected in an unbalanced way. Among the factors affecting the uneven amount of information used in the retrievals include
permanent or temporal obstacles and the time-dependent probability of saturation of the spectra when viewing towards or close to the sun. This means that even if the atmosphere around the measurement site would be perfectly homogeneous in the horizontal plane, the columns retrieved using **V1** and **V2** data sets might be slightly different. We try to explain this by the different sensitivities and their averaging kernels (AK, $AK_{east}^{tot}$, $AK_{west}^{tot}$ and their difference $\Delta AK_{east-west}^{tot}$), through the following equations.

$$x_{ret} - x_{apr} = AK(x_{true} - x_{apr}) + \epsilon \tag{1}$$

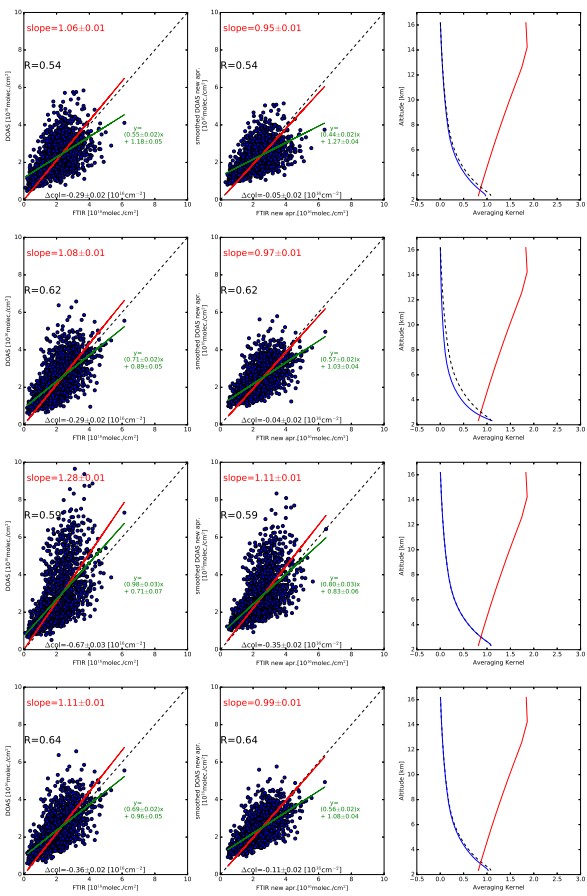

**Figure 5.** Comparison between FTIR and MAX-DOAS measurements conducted at the UNAM measurement site. The first and second row panels correspond to the VCDs retrieved from MAX-DOAS measurements conducted towards the eastern (**V1**) and western (**V2**) measurement sides, respectively. For the third row panel, corresponding to the **V3** data product, the VCDs are retrieved including both measurement sides. The fourth row panel corresponds to the comparison between FTIR and MAX-DOAS **V1** during the morning and FTIR and MAX-DOAS **V2** during the afternoon. The linear regression when forced to zero (red) and not constrained (green) is presented. Black lines represent the 1:1 relation. The left column shows the direct correlation between coincident pairs, whereas the middle column compares the retrieved FTIR VCDs with those calculated from the smoothed MAX-DOAS profiles using the averaging kernel from the FTIR (see text). The right column shows the total column averaging kernel of the FTIR (red lines) and MAX-DOAS (blue lines) retrievals. The dashed black lines on the first, second and fourth row represent the Averaging Kernel of **V3**.

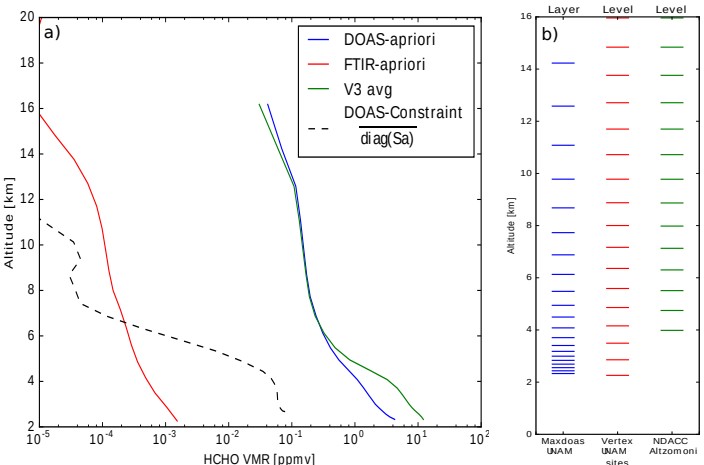

**Figure 6.** a) *A priori* information for the MAX-DOAS and FTIR retrievals. The blue line is the *a priori* used in the MAX-DOAS retrieval, the black dashed line shows the square root of the $S_a$-matrix in the MAX-DOAS retrieval regularized according to optimal estimation (OE). As commonly used for FTIR retrievals that form part of the NDACC network, an *a priori* taken from the WACCAM model was used. The green line is the average of the MAX-DOAS retrieved HCHO profiles (**V3**, both sides) and gives an idea about the vertical distribution of HCHO above UNAM. This averaged profile is used as common *a priori* for the improved intercomparison between MAX-DOAS and FTIR; b) Vertical grids of the MAX-DOAS and FTIR (both measurement sites) inversions.

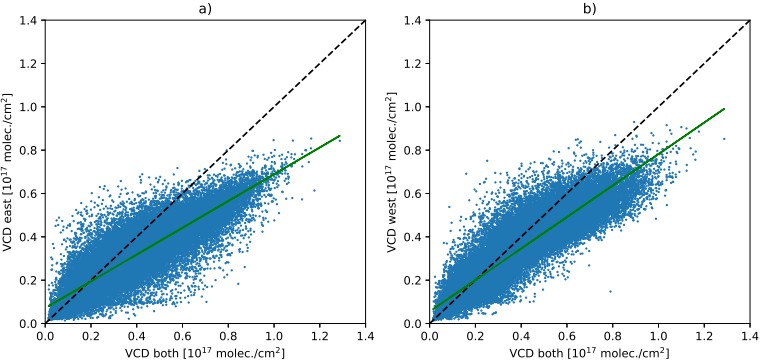

**Figure 7.** Correlation plots between HCHO VCDs retrieved using dSCDs measured towards the (a) east and (b) west with respect to VCDs retrieved using information from both sides (x axis). The green lines indicate the linear regression fits and the black lines represent the 1:1 relation.

Equation 1 (Rodgers, 2000) describes how the retrieved profile is related to the true profile and how other quantities as the total column can be derived. It is described by the following equation, where $g^{tot}$ defines the total column operator for profiles $g_i^{tot} = \frac{\partial col}{\partial x_i}$.

$$col - col_{apr} = g^{tot}(\boldsymbol{x_{ret}} - \boldsymbol{x_{apr}} + \boldsymbol{\epsilon}) = AK^{tot}(x_{true} - \boldsymbol{x_{apr}}) + \epsilon_{col} \tag{2}$$

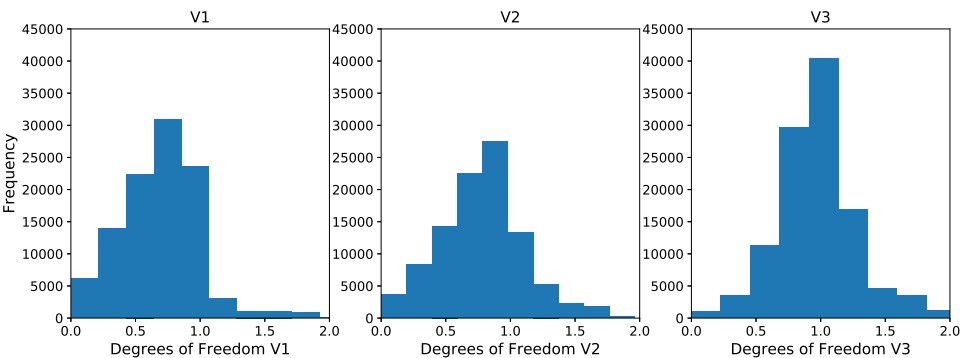

**Figure 8.** Histograms showing degrees of freedom and frequencies for (**V1**), (**V2**) and (**V3**) retrievals.

Here we choose $\boldsymbol{x}$ in the units of partial columns [molec/cm$^2$], so that $AK^{tot}$ is without units as shown in the last column of Figure 5.

$$col_{east} = \quad AK_{east}^{tot}(\boldsymbol{x_{true}} - \boldsymbol{x_{apr}}) + col_{apr} + \epsilon_1 \tag{3}$$

$\quad$
$$col_{west} = \quad AK_{west}^{tot}(\boldsymbol{x_{true}} - \boldsymbol{x_{apr}}) + col_{apr} + \epsilon_2 \tag{4}$$

If the errors $\epsilon_1$ and $\epsilon_2$ of each harmonised retrieval pairs have a random and systematic component, the sistematic component is canceled out for each pair, while the random components result in $\epsilon_{1,2}$ which should be zero in the average of a sufficient large number of measurement pairs.

$$\Delta col_{east-west} = (AK_{east}^{tot} - AK_{west}^{tot})(\boldsymbol{x_{true}} - \boldsymbol{x_{apr}}) + \epsilon_{1,2} \tag{5}$$

If we assume that the retrieved profile $\boldsymbol{x_{v3}}$ using both sides of the measurement plane is to our current knowledge the best estimation for the HCHO profile (if one assumes horizontal homogeneity) $\boldsymbol{x_{true}}$, we use it to simulate the expected differences $\Delta col_{east-west}$.

$\quad$
$$\Delta col_{east-west} \approx \Delta AK_{east-west}^{tot}(\boldsymbol{x_{v3}} - \boldsymbol{x_{apr}}) \tag{6}$$

This expected or simulated difference, which evidently depends on the time of the day, is calculated according to equation 6 and the results are shown in Figure 9 (red line), along with the measured VCDs hourly differences. In this case, the specific distinction is made between the calculated east-west differences of the retrieved VCDs (blue), a simulated VCD difference explained above (red) and the resulting difference between these (calculated-simulated) in black dots.

$\quad$ The averaging kernels from the **V1** and **V2** retrievals, allows us to estimate and forecast a difference, because of their different sensitivities. This effect, dominant after 15 h LT, most likely depends on the number of dSCDs available for the MMF

retrievals that could be significantly reduced as the sun is closer to the viewing angles and do not pass the filtering criteria due to saturation. Alternatively, the forward model in QDOAS could be having more difficulties to explain the measured spectra, so that the errors in the retrieved dSCDs (typically 15% for the lowest elevation angles where the HCHO signal is expected to be higher) could be enhanced and thus unweighted in the MMF model calculations during these afternoon hours due to saturation or the presence of clouds towards the west. The detailed reasons for imbalanced information content between the east and west retrievals are, however, not yet investigated. Nevertheless, the calculation of the red line uses the HCHO profile retrieved in the **V3**, as it is probably the best estimation available, but it is of course not the true profile and the meaningfulness of equation 6 is therefore limited to be qualitative. Without grouping in different hours of the day, both differences originated from the gradient and from the information content cancel partly out and would not show a conclusive result.

As can be seen in Figure 9, the simulated line resembles the calculated VCD differences (blue line) and both lines present higher positive differences in the morning hours and a decline towards negative values in the afternoon and hence indicate that a large part of the observed difference is due to differences in information content and not necessarily due to different real distributions. However, the difference between the calculated and simulated lines, shown as black dots, gives us a better indication of the relative HCHO abundances along the east-west viewing direction. During the morning hours, this calculated difference has positive values demostrating that the abundance of HCHO on the eastern side of the scanning plane is higher than in the western side. After 12 h LT, conditions change so that larger HCHO VCDs are measured towards the western side of the scanning plane, peaking at 13-14 h. The average HCHO distribution over the MCMA, reconstructed from OMI data (Figure 4), provides evidence of a larger enhancement of HCHO columns towards the western side of the MCMA at OMI overpass time, coinciding with our findings in terms of the identified horizontal inhomogeneity as well as timing. Afterwards, the situation appears to slightly return to the circumstances where HCHO is more evenly distributed in both directions and might be even a bit higher towards the east at around 16 h. The observed changes could also be related to orographic and meteorological conditions. Fast et al. (2007) report that surface wind measurements over the MCMA indicate the production of strong convergence in the basin during the late afternoon, created by opposing propagating density currents and a gap flow originating in the southeastern corner of the basin. The authors conclude that in the MCMA short-range transport can be produced by the complex terrain surrounding it, producing local and regional circulations.

### 3.3.3 Time-dependent MAX-DOAS vs FTIR comparison

In the previous section, it was shown that analyzing the MAX-DOAS viewing directions independently can in part explain the large horizontal inhomogenity around the UNAM urban site. We now investigate the behaviour in the correlation between MAX-DOAS and FTIR data for different hours of the day and how it can be affected by the sensitivities of both instruments and the changing vertical distribution of the HCHO profiles.

Figure 10 shows the comparison of hourly MAX-DOAS (**V3**) vs. FTIR measurements at UNAM. For each hour the correlation coefficient is provided, along with the linear regression information when forced to zero (red) and not constrained (green). Black lines represent the 1:1 relation. In the lower right panel, the FTIR and MAX-DOAS averaging kernels as a function of altitude are plotted for each hour, not showing significant variability. Hourly correlation coefficients range between 0.71 at 9

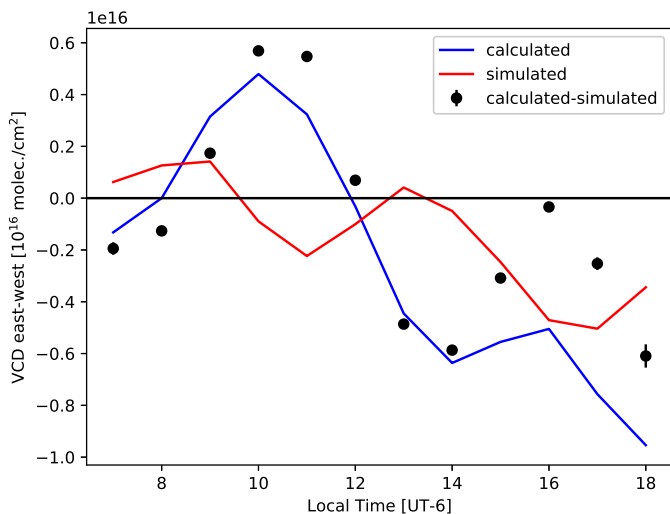

**Figure 9.** Average differences between the retrieved columns using the east and west measurement sides, as function of the hour of the day, are presented in the blue line. The red line simulates how the information available from the retrievals would produce a difference in the columns, assuming that the retrieval of the **V3** data set using both measurement sides best describes the atmospheric state (see text for details). The black points is the difference between the found differences and the "forecasted" differences, and should show the part of the difference which might be related to a real inhomogeneity and a gradient between the east and the west HCHO concentrations in the mixing layer. The error bars represent the standard error and show that the measurement amount is large enough to calculate a statistically significant mean difference for each day hour, and the grouping for different hours is necessary.

h LT and 0.49 at 13 h, while slopes vary between 1.62 (12 h) and 0.93 (15-16 h). It is interesting to point out that the slope steadily increases between 9 h (1.21) and noon (1.62), the latter being the precise inflection point where the MAX-DOAS instrument reports a significant change in the HCHO VCDs horizontal distribution (Figure 9). At 13 h (1.45) the slope starts to

5    decrease until it stabilizes at 15-16 h (0.93). On the other hand, the correlation coefficient steadily decreases between 9 hours (0.71) and 13 h (0.49), increasing afterwards, reaching a final value of 0.6 at 16 h.

The relation between the MAX-DOAS and the FTIR VCDs is described by the scatter (the Pearson correlation coefficient), the slope and constant bias. As we already have seen in the comparison between the **V3** MAX-DOAS data product (both sides) with respect to the single sides, having a slope of 1.0 does not ensure that both retrievals are correct and similar to the true

10   atmospheric state, but it rather means that both sensitivities are similar.

Based on Equation 1 (Rodgers, 2000) and assuming that the variability in the HCHO concentration profile can be described by a Gaussian probability distribution with the mean profile ($x_{mean}$) and a covariance matrix $Sa$ (unfortunately not known), the Pearson's correlation and the slope in the scatter plot of two retrievals can be calculated using the averaging kernels and the errors of both retrievals.

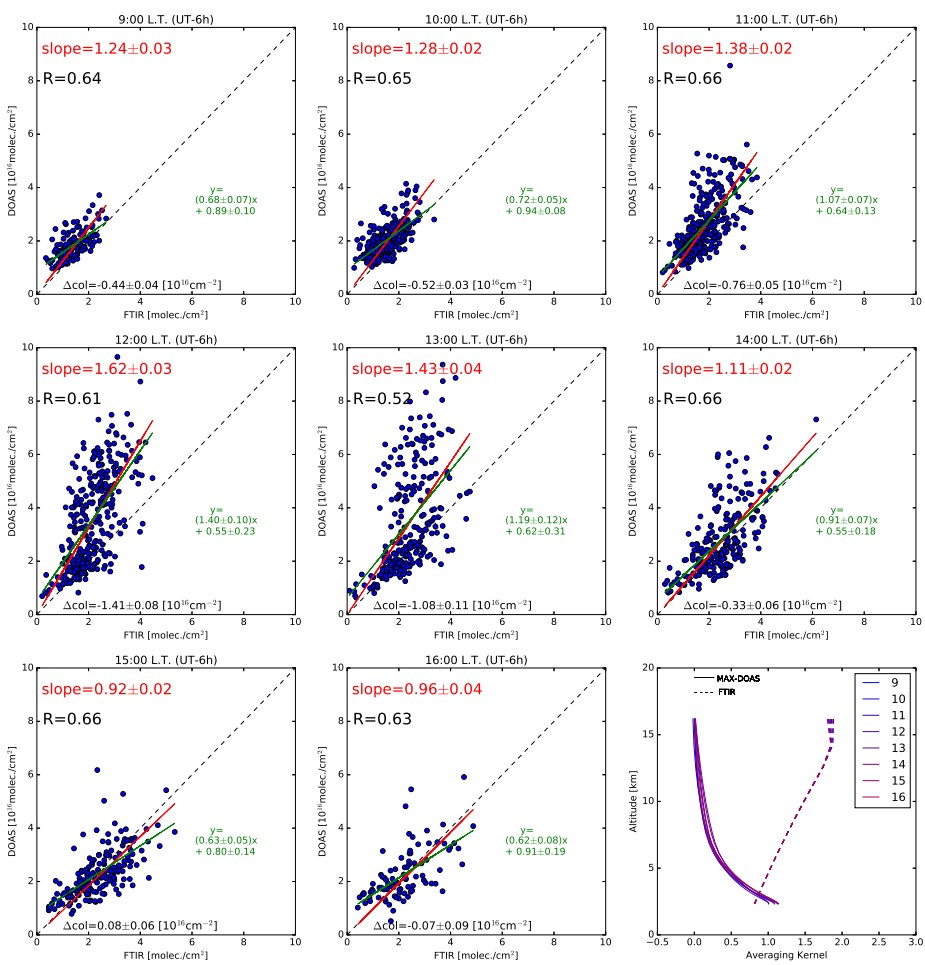

**Figure 10.** Hourly FTIR vs MAX-DOAS comparisons at UNAM between 9 and 16 hours (local time).

Neither the retrieved FTIR profile (1.1) nor the MAX-DOAS profile retrieval (<2) have sufficient degrees of freedom to consider the retrieval as profile retrieval, therefore the strategy of using the profile information from one instrument together with the averaging kernel of the other instrument is not too promising.

5  Here we try to evaluate the consistency of the two retrievals differently. Starting with Equation 2 and assuming that both instruments are measuring coincidently the same atmospheric state $x_{true}$.

The average of the product of the columns of both instruments is theoretically given by the following equation, for that purpose we introduce the errors $\epsilon_{FTIR}(i)$, and $\epsilon_{DOAS}(i)$, where $i$ is the index which identifies a certain hour:

$$\langle \Delta col_{DOAS} \cdot \Delta col_{FTIR} \rangle = \frac{1}{N} \sum_{i}^{N} [AK_{DOAS}^{tot}(i)(\boldsymbol{x}_{true}(i) - \boldsymbol{x}_{apr}) + \epsilon_{DOAS}(i)] \cdot [AK_{FTIR}^{tot}(i)(\boldsymbol{x}_{true}(i) - \boldsymbol{x}_{apr}) + \epsilon_{FTIR}(i)] \tag{7}$$

10  To simplify the interpretation we assume that the Averaging Kernels of both instruments are more or less constant and independent in $i$, where $i$ is the index for the hours for which coincident measurements exist. This assumption is valid for the FTIR-Averaging Kernel (Figure 10, Equation 7), but it is not the case for the MAX-DOAS measurements and it is therefore an approximation, which might be verified in each case. Another simplification is the assumption that, the retrievals are corrected using the same apriori $x_{apr}$ profile, which is also the mean of the true profiles $x_{true}$ of the ensembles. So the average of 15 $\Delta col_{DOAS}$ and $\Delta col_{FTIR}$ are zero. In addition we assume that the errors $\epsilon_{FTIR}(i)$, $\epsilon_{DOAS}(i)$ are independent and in average zero, we assume also that they are independent with respect to $AK_{DOAS}^{tot}(i)$, $AK_{FTIR}^{tot}(i)$, $\boldsymbol{x}_{true}(i)$, so that we can simplify the calculation of equation 7 to equation 8.

$$\frac{1}{N} \sum_{i}^{N} AK_{DOAS}^{tot} \underbrace{[(\boldsymbol{x}_{true}(i) - \boldsymbol{x}_{apr}) \cdot (\boldsymbol{x}_{true}(i) - \boldsymbol{x}_{apr})^{T}]}_{Sa} \cdot (AK_{FTIR}^{tot})^{T} \tag{8}$$

and therefore the Pearson correlation coefficient and the slope are formally calculated:

$$20 \quad R_{pearson} = \frac{(AK_{DOAS}^{tot}) \cdot Sa \cdot (AK_{FTIR}^{tot})^{T}}{\sqrt{((AK_{DOAS}^{tot}) \cdot Sa \cdot (AK_{DOAS}^{tot})^{T} + \sigma_{DOAS}^{2})((AK_{FTIR}^{tot}) \cdot Sa \cdot (AK_{FTIR}^{tot})^{T} + \sigma_{FTIR}^{2})}} \tag{9}$$

$$slope = \frac{(AK_{DOAS}^{tot}) \cdot Sa \cdot (AK_{FTIR}^{tot})^{T}}{(AK_{FTIR}^{tot}) \cdot Sa \cdot (AK_{FTIR}^{tot})^{T}} \tag{10}$$

If the correlation plot is limited to just one hour as shown in Figure 10, the $Sa$ just describes the variability in that hour. For 25 a certain time of the day it is more probable that the variability and covariance matrix $Sa$ is described by a single dominant Eigenvector $Sa = \boldsymbol{v} \cdot \boldsymbol{v}^{T}$. If the errors $\sigma_{DOAS}$ and $\sigma_{FTIR}$ in the columns of MAX-DOAS and FTIR can be neglected, the resulting Pearson's correlation coefficient would be 1.0 according to equation 9 and the slope would be given by $(AK_{DOAS}^{tot} \cdot \boldsymbol{v})/(AK_{FTIR}^{tot} \cdot \boldsymbol{v})$, the quotient of the weighted averaged total column averaging kernel elements using the weights $\boldsymbol{v}$.

The variability of the concentration profile with a fixed shape ($v = \lambda_v e_v$, with $\lambda_v = |v|$) from one day to another, has a strong impact on the Pearson's correlation coefficient R (more variation with respect to the errors (FTIR and MAX-DOAS) results in a better R-value), but not on the value of the slope. The slope is given by the averaging kernels of the two instruments and the shape of the variable profile $v$. In Mexico City, we could assume that at 9 h LT the mixing layer is well mixed with HCHO up to a certain height with a constant concentration but with 0 or at least a constant HCHO value above this height. For this simple assumption (the only Eigenvector is constant in the mixing layer but 0 above it), the slope is the fraction of the mean averaging kernel elements in the mixing layer (MAX-DOAS/FTIR). In case we cannot explain the experimental measured slopes, we learn that there are some other processes involved, which are not described by the simplified $Sa$. Maybe there might be sometimes also a pollution plume above this well mixed layer.

The individual plots in Figure 10, showing the correlations and their slopes for each hour, allow us to support the fact that instead of simply cross validating the FTIR and MAX-DOAS retrievals, it is possible to assume that the mixing layer height dominates the variability and that such simplification is valid on a certain hour. The validation is therefore given by the fact that a plausible variability for each hour explains the slope and correlation for different hours, rather than that the slope and the correlation is close to 1.0.

### 3.4 Background HCHO variability in a remote site

In order to investigate background HCHO levels, HCHO VCDs retrieved from measurements conducted with the high resolution FTIR spectrometer (Bruker HR120/5) in Altzomoni are presented in Figure 11. HCHO VCDs measured at Altzomoni are in the same order of magnitude as HCHO VCDs reported by Vigouroux et al. (2018) for several "clean" sites stations belonging to the NDACC network, such as Zugspitze, other mountain site (however at a latitude of 47°and an altitude of 3 km) as well as for Mauna Loa, at a latitude of 20°and an altitude of 3.4 km.

As in the case of the UNAM measurement site, the diurnal and seasonal HCHO cycles were calculated. Hourly HCHO VCDs at Altzomoni show a steady increase during the day, with a smaller growth-rate from 14 to 17 h. Vigouroux et al. (2018) report the same behavior (a maximum in the late afternoon between 16 and 18 local time) for other stations of the NDACC network: Bremen, Paris, Toronto and Lauder. Further analysis should be conducted regarding the diurnal HCHO cycle at Altzomoni, however the detected maximum at late afternoon could be attributed to upslope transport or to secondary HCHO production that has reached a maximum at a certain hour of the day.

The seasonal cycle shows at maximum during September, while the lowest HCHO VCDs values occurr during December. The background HCHO VCDs at Altzomoni (Figure 11) are an order of magnitude lower than the values reported at the urban UNAM measurement site for both the diurnal (Figure 2) and seasonal (Figure 3) cycles.

## 4 Discussion & Conclusions

In this contribution we present a comparison between HCHO total column densities retrieved from two independent measurement techniques: MAX-DOAS and solar absorption FTIR. Both measurement techniques, although based on spectroscopy,

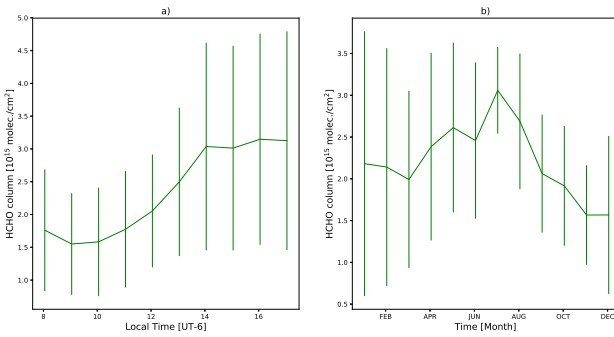

**Figure 11.** Hourly (a) and seasonal (b) HCHO VCD cycles at Altzomoni from FTIR measurements.

exhibit a very different measurement strategy and geometry. Albeit these differences, a good agreement was obtained between both instruments. Due to the versatility of the retrieval code used to process the MAX-DOAS data, VCDs were retrieved using measurements conducted towards different viewing directions. Retrieval products were obtained employing measurements conducted exclusively towards the east, the west, or using measurements conducted towards both sides of the measurement station. Considering the FTIR results as the reference, MAX-DOAS VCDs from these data sets where 5%, 9% and 28% larger than FTIR, respectively.

Reasons for the overestimation of the MAX-DOAS over the FTIR results are attributed to an enhanced ground level (lowest few kilometers of the atmosphere) sensitivity of the former with respect to the latter. However, the intrinsic differences between the two measurement techniques could also account for the discrepancies found in this study. In the first place, both measurement techniques have different sampling geometries and strategies. The MAX-DOAS instrument measures spectra at different elevation angles, leading to an altitude-averaged measurement in the lower atmosphere. From these measurements, HCHO VCDs are computed using a numerical code -MMF- (Friedrich et al., 2019). For the MAX-DOAS instrument, typically, a complete cycle encompassing several observations may take several minutes, which could be a disadvantage during periods of rapidly varying radiation transport conditions in the atmosphere, such as varying cloud cover, aerosol load, or during sunrise and sunset. The distance from which photons are scattered and reach the instrument's telescope is variable, and depending on atmospheric conditions and the wavelength it has been calculated to be between 0.6 and 6 km (Platt and Stutz, 2008). Meanwhile, the FTIR instrument receives direct sunlight, so that the air mass sampled by the instrument comes from a well defined cone angle smaller than the solar disc (external field of view around 7 mrad at UNAM and 2 mrad at Altzomoni) and the time resolution of the FTIR-spectra are described by the measurement duration and frequency. The measurement duration and time distance are 1 min duration and every 5 minutes at UNAM and 7 min duration and every 21 minutes at Altzomoni. After computations, direct sun total HCHO column is delivered. Further characterization of the differences between both measurement techniques, for example by using the same sampling geometry, is work in progress for the Spectroscopy and Remote Sensing Group at CCA-UNAM and in the future will provide further insights about the differences and possible biases between the two

techniques due to distinctive spectroscopic characteristics and retrieval approaches. In addition, a different analysis approach could be taken in the future, by using and presenting both data sets (FTIR and MAX-DOAS) as complementary to each other, since due to the different sensitivities of the measurement techniques, the retrieved information do not refer exactly to the same altitudes of the atmosphere.

Moreover, this research provided the opportunity to study in more detail horizontal HCHO inhomogeneities in the MCMA, identifying diurnal and seasonal variabilities of the abundance of HCHO total columns which in the future could be used to further study primary vs. secondary HCHO in the MCMA and develop specific analysis strategies focused on the identification and disaggregation of freshly emitted and secondary produced HCHO in the boundary layer of the MCMA. Satellite-based data has been used to corroborate the spatial inhomogeneity of the HCHO total column over the MCMA as shown in Figure 4, strengthening the importance to continue these type of inhomogeneity studies at different azimuthal angles, in different zones of the MCMA and also focusing on other atmospheric constituents. The identified inhomogeneity of HCHO in Mexico City could be investigated even further by using the lowest elevation angles of the MAX-DOAS data (i.e. near-surface HCHO) at the different azimuth angles.

Identifying and characterizing horizontal inhomogeneities with respect to the abundance of molecules present in air can also be of service when making decisions regarding location and azimuth measurement angles for future MAX-DOAS stations in the MCMA. Future work includes to study horizontal inhomogeneities at other stations of the MAX-DOAS network as well as horizontal inhomogeneities of other chemical species, such as nitrogen dioxide, which is routinely retrieved as well from the spectra measured by the MAX-DOAS instruments located in the MCMA.

It is worth mentioning that these type of strong spatial heterogeneity scenarios have been observed in different areas of the planet and specific studies of atmospheric constituents have been or are currently being performed in other urbanized and densely populated areas such as North America (Boeke et al., 2011; Chance et al., 2000; Millet et al., 2008), China (Cheng et al., 2015; Zhang et al., 2019), particularly in the Beijing-Tianjin-Hebei region (Zhu et al., 2018) and the Yangtze River Delta area (Chan et al., 2019; Hong et al., 2018; Tian et al., 2018; Wang et al., 2016); South Asia (Rana et al., 2019) and more specifically India (Chutia et al., 2019; Surl et al., 2018) and Pakistan (Khan et al., 2018; Khokhar et al., 2015). In addition specific case studies have been conducted globally (Wittrock et al., 2006) or in the Southern Hemisphere (Ahn et al., 2019), the Atlantic Ocean (Behrens et al., 2019) and the East China Sea (Tan et al., 2018). Findings of these studies imply enhanced abundance of HCHO over highly populated areas, areas with increased industrial activity, zones exhibiting biogenic emissions and biomass burning activities, along major highways and in some instances identifying cases of regional transport of pollutants.

The quantified diurnal and seasonal variability of HCHO as well as the characterized horizontal inhomogeneity in the MCMA, presented in this study, could be attributed to direct emissions or secondary formation of HCHO from released precursors from anthropogenic and/or biogenic sources that form part of the MCMA and constantly influence its atmospheric composition. Identification of either primary emissions or secondary formation of HCHO is outside the scope of this study, however, based on the results presented here and previous research conducted by Garcia et al. (2006), future analyses could include studying other molecules present in the atmosphere as tracers of primary or secondary HCHO.

In terms of further characterizing HCHO horizontal inhomogeneity, the Tropospheric Emissions: Monitoring of Pollution (TEMPO) instrument (Zoogman et al., 2017), an airborne mission to be launched in the near future, will allow us to corroborate the hourly horizontal changes in the HCHO distribution over the HCHO that have been identified in this study.

*Data availability.* The MAX-DOAS and FTIR data can be accessed via http://www.epr.atmosfera.unam.mx/maxdoas_data/hcho/ and http://www.epr.atmosfera.unam.mx/ftir_data/hcho/ respectively (last access: 7 May 2020). Data from other stations, with other versions or periods measured within Mexico City's MAX-DOAS network, should be requested from Claudia Rivera (claudia.rivera@atmosfera.unam.mx).

*Author contributions.* CR is responsible for the QDOAS retrieval setup and parameter choices, for the setup and running of the MMF code, for processing HCHO OMI data as well as for data analysis and interpretation. CR wrote parts of the Abstract and parts of Sect. 2, 3 and 4. CG is responsible for developing and optimizing various HCHO retrievals for Altzomoni and UNAM. Contributed to the developement of the NDACC retrieval strategy which was finally applied. CG has taken care of the measurements and focused in the separation of fresh emitted HCHO and secondary produced HCHO in the boundary layer of Mexico City. WS wrote parts of Sect. 2 and 3. MMF is responsible for the retrieval MMF code development and testing, the retrieval chain setup from spectra to profiles, and the retrieval parameter choices for MMF and software support. MMF wrote parts of Sect. 2. DR is responsible for running parts of the MMF code. CAMR is responsible for processing HCHO OMI data and assisting on creating Fig. 1. JA and AB provided technical support for instruments and data management. MG was involved in the data interpretation and wrote Sect. 1 and parts of section 2, 3 and 4. WS, AB and MG are responsible for running the FTIR retrievals at UNAM and Altzomoni and for data analysis and interpretation. TB and FH provided the HR 120/5 spectrometer located in Altzomoni and developed the setup of the spectrometer and solar tracker. Provided assistance to take the spectrometer in Altzomoni into operation and trained the Mexican group in the operation and alignment of the HR120/5. FH developed the retrieval code PROFFIT9 and LINEFIT and provides continous support on its use. In terms of specific figures contributions, Figs. 1, 2 and 3: WS and CR; Fig. 4: CR and CAMR; Figs. 5 and 6: WS; Figs. 7, 8 and 9: WS, DR and CR; Figs. 10 and 11: WS.

*Competing interests.* The authors declare that they have no conflict of interest.

*Acknowledgements.* This research has been supported by the DGAPA-UNAM (grants no. TA100418, IN111418, IN107417 and IA101620), the CONACYT (grant no. 290589), and the INECC (grant INECC-A1-002-2019). C.A.M.R acknowledges financial support from Consejo Nacional de Ciencia y Tecnología (CONACYT) through a graduate studies grant, CVU 956921. Arne Krueger, Josué Arellano, Alfredo Rodríguez, Delibes Flores, Miguel Angel Robles and Omar López are thanked for their technical assistance. We thank Caroline Fayt (caroline.fayt@aeronomie.be), Michel Van Roozendael (michelv@aeronomie.be) and Thomas Danckaert for the free use of the QDOAS software and we thank Robert Spurr for free use of the VLIDORT radiative transfer code package. We thank Agustin García Reynoso for impotant and fruitful discussions about HCHO in Mexico City. We thank the Mexican Solarimetric Service for their effort in establishing and maintaining

the AERONET Mexico City site. We thank the University of Wyoming Department of Atmospheric Science for providing the sounding data on http://weather.uwyo.edu/upperair/sounding.html (last access: 29 April 2020).

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
