# Peer review of "Formaldehyde total column densities over Mexico City: comparison between MAX-DOAS and solar absorption FTIR measurements"

_Atmospheric Measurement Techniques, 2020_

## Referee Comment (RC1) · Anonymous Referee #1 · 28 Sep 2020

Review of Rivera et al. (2020): Formaldehyde total column densities over Mexico City: comparison between MAX-DOAS and solar absorption FTIR measurements

The manuscript presents 6 years of formaldehyde vertical columns VCD measured from Mexico using two different techniques: FTIR and MAXDOAS. The MAXDOAS dataset comprises 3 different type of observation geometries depending on the azimuth of the instrument (V1: East, V2: West and V3: mixed V1 and V2). The comparison of the results obtained with the different techniques shows that, in general, the total columns retrieved with the DOAS instrument are higher than those from the FTIR VCD (V1: 5% higher than FTIR, V2: 9% higher and V3: 28% higher than FTIR). The authors

conclude that the differences relay on the different sensitivity of each technique and also on the horizontal distribution of HCHO in the troposphere.

In general, the discussions are mainly focused on the comparison of V3 with FTIR, and the internal consistency between the V1, V2, and V3 data. Moreover, supported by satellite HCHO VCD, through these data the authors point out the horizontal heterogeneity of the HCHO distribution in Mexico City. This confirms, once more, the relevance of ground-based observations for satellite validation and it also emphasizes that, for satellite validations, scientists should bear in mind not only the different sensitivities of the different techniques and observation geometries, but also the effect of horizontal inhomogeneity in the distribution of trace gases.

In addition to the urban HCHO VCD data, the authors also present HCHO VCD data observed with a FTIR located in the high-altitude site of Altzomoni.

The manuscript is well presented and the methodology is well described. Moreover, the MAXDOAS and FTIR data of Mexico City and the FTIR data of Altzomoni presented in the manuscript (6 years) are quite unique and may serve for future publications on more detailed chemical/transport processes in such megacity.

Overall, publication is recommended after addressing the following comments.

—————————General Comments————————-

Beyond Sect. 3.3.1, Sect. 3.3.2 and Sec. 3.3.3, one simplistic view of investigating the gathered data would be that, since FTIR performs direct Sun observations and DOAS V1 sense air masses towards the East, and DOAS V2 towards the West, the FTIR observations could be split between morning and afternoon observations in order to compare to V1 and V2 DOAS data, respectively. For instance, once smoothed by the averaging kernel, have the authors compared the FTIR morning data to morning V1 DOAS, and FTIR afternoon data to afternoon V2 DOAS? Would that compare better (or worse) to the FTIR am/pm data than V3? Also, although it is mentioned throughout

the manuscript, it is relevant that the authors state "up front" that, due to the different sensitivities of the techniques, the FTIR and the MAXDOAS information do not refer exactly to the same altitudes of the atmosphere (e.g., averaging kernel in Fig. 5). More than comparing one data set to the other, it might be more useful to use (and present) both data sets as complementary to each other. Also, the inhomogeneity of HCHO in Mexico City could be investigated even further by using the lowest elevation angles of the MAXDOAS data (i.e., near-surface HCHO) at the different azimuth angles (although that might result in another paper by itself).

Also, in addition to the urban data, this work presents HCHO observations form the high-altitude site of Altzomoni (FTIR). Given the sparse number of measurements of HCHO performed from high-altitude locations, the manuscript would probably benefit if the authors could dig a bit further on these data set since, in fact, these data are merely presented in one small paragraph in the manuscript (Sec. 3.4.), but not really discussed (e.g., how do the HCHO VCD in clean and in urban scenario compare or the reasons behind the daily and seasonal evolution shown in Fig. 11), or put in context (even if briefly) with HCHO observations from other high altitude sites worldwide.

————————Specific Comments—————————

P1, L11-12: "A time-dependent comparison revealed that the vertical distribution of this pollutant, guided by the evolution of the mixing layer height, can play an important role in how the results are affected."

Do the authors mean that the vertical distribution of HCHO can play an important role in how the results are affected? i.e., are the results affected by the distribution of HCHO? Please, clarify.

P1, L21: HCHO is mainly (not "also") formed from the oxidation of CH4 and NMVOCs. This is particularly relevant for the Altzomoni data presented.

P4, L27: How long does it take to perform 1 scan (i.e., 90, 0, 2, 6, 13, 23, 36, 50, 65,

82)?

P4, L31: In addition to the mentioned filters, is any sort of cloud filter applied to the DOAS data?

Please, similarly to the error estimation offered for FTIR observations (P4, L 14), provide an error estimation of the MAXDOAS data presented in the manuscript.

P5, L2: 324.6-359 nm is chosen for the HCHO spectral DOAS analysis. Why that particular range instead of the one suggested by e.g. Pinardi et al (2013)? Given the chosen spectral range, the possible spectral interference of BrO and/or O3 should be addressed (maybe a test with chosen days?) not only for the city data but also for the high-altitude observations. Also, what polynomial did the authors use for the HCHO DOAS retrieval? Note the impact of the polynomial mentioned by Pinardi et al. (2013).

P6, Sec. 3.1: Since Section 2 addressed FTIR and MAXDOAS observations,, and Section 3 is entitled as "Results", probably Sect. 3.1 would make more sense after presenting the results of FTIR and MAXDOAS observations (Sect. 3.2).

P6, L14: why is V3 chosen (and not V1 and/or V2)? Since V1 and V2 are referred throughout the manuscript, the authors may want to include the time series of not only the V3 VCD in a figure (Fig. 2), but also of V1 and V2. Also, how is V3 retrieved? i.e., do the authors averaged the dSCDs observed at V1 and V2, and then invert V3 VCD from that averaged V1+V2 dSCD?

P7, figure 2: Based on the averaging kernels shown later in Fig. 5, it would be helpful to remind the reader the altitude ranges covered by each instrument (e.g., FTIR UNAM VCD 2-16 km; FTIR UNAM VCD 4-16 km; MAXDOAS UNAM VCD 2-5 km)

P7, L4: "…and thus probes cleaner atmospheric columns" as long as there is no upslope transport (is there at Altzomoni?). Also, how do these VCD at Altzomoni compare to HCHO observations at other high altitude research sites? As mentioned in the general comments, authors are kindly advised to address further the results of Altzomoni

data throughout the manuscript since those data are relevant by themselves (note the very scarce HCHO observations from high-altitude sites).

P7, L6: "…in general larger than…". How much larger? Please, quantify.

P7, L13: "…VCDs are larger than…". How much larger? Please, quantify.

P9, Sect. 3.1.1: See general comments (i.e., are the FTIR morning data comparable to morning V1 DOAS, and FTIR afternoon data comparable to afternoon V2 DOAS?)

P9, L24: Please, specify (i.e., quantity) the (average) degrees of freedom (DOF) of the retrieved VCD for each technique (not only MAXDOAS but also FTIR). As for the DOAS V1, V2, V3 measurements, do they have similar degrees of freedom? Figure 8 shows they are not the same, please provide an average DOF for V1, for V2 and for V3 or the time series.

P10, Fig. 5: how is the vertical grid of the FTIR inversion compared to the one used for the MAXDOAS inversion?

P13, L8-9: "the retrieved profile using both sides of the measurement plane is to our current knowledge the best estimation for the HCHO profile" if one assumes horizontal homogeneity (?).

P13, L19: please provide a DOAS dSCD error estimation.

P15, L21: "…both instruments are measuring coincidently the same atmospheric state" Would that be true given Fig. 1? Probably only if V3 is used as measuring vector (?)

————————Technical Corrections————

P1, L 3-6: For the MAX-DOAS measurements, the software QDOAS was used to calculate differential Slant Column Densities (dSCDs) from the measured spectra and subsequently the Mexican MAX-DOAS Fit retrieval 5 code (MMF) to convert from dSCDs to Vertical Column Densities (VCDs). The direct-solar absorption spectra measured

with FTIR were analyzed using the PROFFIT retrieval code These sort of details would be better if included later in the text (Sec. 2.1, 2.2), not in the abstract.

P1, L 10: "could demonstrate"

Probably more accurate would be "suggests" or "indicates"

P1, L 12: "Apart from the reported…"

In addition to the reported…

P2, L 12: "… satellites, aircraft, vehicles or ground based"

Balloons as well

P2, L 20: Since CINDI is mentioned, probably the authors should also mention the more CINDI2 campaign (Kreher et al., 2020)

P2, L26: LP in LP-DOAS was not introduced before (i.e., long-path DOAS)

P2, L24-28: FTIR vs MAXDOAs literature. The authors may want to consider including Franco et al., AMT, 2015;

P2, L30: "Another study by Garcia et al. ….dominate the HCHO concentration at the surface". The authors may want to split that long sentence.

P3, L5: In which sense is the work presented an "unprecedented comparison"? Is it due to the length of the study (i.e., 6 years)? Is it due to the location of the study (i.e., Mexico)?

P3, L7: "to characterize the difference" in?

P3, L5-12: To ease the reader, the authors may want to specify in which section will be addressed each of the topics mentioned in this paragraph.

P3, L15: "One" of the sites "is…"

P3, L17: "The other" site "is the Altzomoni…"

P3, L21: At Altzomoni, please specify that the FTIR is part of NDACC. Note that NDACC also comprises DOAS instruments.

P5, L4: Even if it is mentioned by Friedrich et al., to ease the reader the authors may want to include at least the spectral range where O4 is retrieved.

P6, Fig. 1: Please, enlarge the size letter of the two sites in the map, they are hard to see. Also, a circle indicating the region that comprises MCMA would be helpful for the reader.

P7, L11: "the seasonal HCHO VCDs" Are those the monthly average data?

P7, L12: The meaning of the black line is not clear. Also, does it relate to FTIR or to MAXDOAS? Note that in Fig. 4 the black data are very hard to distinguish.

P13, L13: What do the authors mean with "the calculation of the red trace"? Do they mean "line"? Note that the equations in page 13 refer to matrices and the word "trace" might be misleading. If refer to line, authors are advised also to change it in the caption of figure 9.

P14, Fig. 9: A horizontal line at VCD difference = 0 might help the reader to understand that figure.

P15, L25: "i" stands for?

P15, Eq 8: Equation incomplete

P17, L15-18: "The slope is given by the averaging kernels of the two instruments and the shape of the variable profile v, and for the simple assumption described above, that the only Eigenvector is constant in the mixing layer but 0 above it, the slope is the fraction of the mean averaging kernel elements in the mixing layer (MAXDOAS/FTIR)."

The authors may want to split this very long sentence.

P17, L21: Given all the assumptions needed, more than "to demonstrate", probably it

would be better "to support"

P17, L21-24: The paragraph is a bit confusing. The authors may want to clarify what they mean.

P18, L20: "...depending on atmospheric conditions ..." and the wavelength.

P19, L28-36: This paragraph presenting the megacity of Mexico might fit better in the introduction (Sect. 1).

---

## Referee Comment (RC2) · Anonymous Referee #2 · 30 Sep 2020

The paper presents MAXDOAS and FTIR HCHO measurements around Mexico City, for a period of about 6 years. MAXDOAS data at several viewing directions are included, showing horizontal inhomogeneities. Comparisons between the MAXDOAS data and FTIR are presented, resulting in MAXDOAS larger values, from about 5% to 28% depending on the direction. An FTIR background site is also briefly presented, with its daily and seasonal variations. The scientific content of the paper fits well the scope of AMT and the manuscript is well written and of interest for the community. I recommend publication after the suggested revisions below.

General comments

[Figure]

The datasets are important (in length and for their high HCHO VCD columns) and the comparison of MAXDOAS and FTIR is of great interest, and they would deserve some more explanations. FTIR datasets have already been used in other publications (e.g., Vigouroux et al., 2018; 2020) and this should be emphasized a bit more, putting the 2 sites in the context of other existing HCHO FTIR. Also the Altozomoni site is showed in Sect. 3.4, but only very shortly.

MAXDOAS HCHO data from Mexico is presented here for the first time (to my knowledge), and these datasets (v1, v2 and especially v3) need a bit more explanations. In Sect. 2.2, the MAXDOAS error estimation are not even mentioned! Information on the polynomial and offset choice in Table 1 are missing, and more explanations of the v3 analysis should be given. It is said that "VCDs retrieved using measurements from both sides of the scanning plane are in general larger than VCDs retrieved using data from measurements of only one of the sides. This result can be explained by the larger amount of information available for the retrievals when dSCDs in different elevation angles and both scanning directions are used" (end of P.9 and P. 11), but it is never mentioned how this v3 is done. How are the opposite directions treated in term of a-priori, aerosols content, …? Is the retrieval considering an homogeneous atmosphere for the retrieval, or are the differences for v1 and v2 somehow taken into account for the v3 retrieval? An illustration of the behaviors of v1, v2 and v3 on a typical day would be a nice addition (and also adding v1 and v2 on the diurnal and seasonal figures 3 and 4). Also the degrees of freedom should be quantified (numbers in figure 8 are small and difficult to read).

Add reference and discussion of Vigouroux et al., 2009 (Reunion Island), and Franco et al., 2015 (Jungfraujoch) to better highlight the difference in sensitivities between MAXDOAS and FTIR. The AVK are shown in the last column of figure 5, but they maybe deserve a specific figure instead, comparing the AVK of FTIR, MAXDOAS v1, v2 and v3 on the same panel. When seeing the AVK, the 2 measurements are not sensitive at all at the same altitudes, so if the HCHO is not homogeneously distributed,

we don't expect the same measurements.

The comparisons between MAXDOAS and FTIR are a bit perturbing, as Figures 2 to 4 seems to suggest a bias of 50%, when all the data are considered (are the MAX-DOAS data cloud filtered?), while figure 5, when the coincident measurements are selected, seems to indicate smaller bias (28% if considering v3 with the slope passing by the origin – although the non-forced regression show an important systematic intercept). The different regressions of Fig. 5, should be discussed in more details. Also figure 2 shows better MAX-DOAS to FTIR comparison during 2013-2014, while since 2015 the MAX-DOAS are up to twice the FTIR values. Is there a reason for that? Has the instrument or the measurement strategy changed?

Instead of (or in addition to) comparing hourly MAXDOAS v3 to FTIR in Figure 10, why not compare the morning FTIR data with the MAXDOAS measuring to the East, and afternoon FTIR to MAXDOAS measuring to the West? Adding the measurement directions to the map of Figure 1 could help the reader understanding the measurements at each site. Giving some explanations on the inhomogeneities in HCHO seen by OMI could also help the reader (is there any specific vegetation? Industry? How is the orography around Mexico city? Can some HCHO be "trapped" by winds and terrain?). Are the conclusion of Section 3.3.2, with the larger abundances on the eastern side of the scanning plane during the morning hours, and a change after 12h LT, supported e.g. by wind direction changes? How is this gradient explained? Are the other MAX-DOAS Mexican sites HCHO measurements confirming this horizontal inhomogeneities?

Also, a more fundamental question. There has been recent studies (e.g. with Pandora, https://amt.copernicus.org/preprints/amt-2020-158/) showing contamination of "plastic" material from the instrument, emitting HCHO in case of hot temperatures – is this eventuality been excluded here? Is there any relation of the measured HCHO with the temperature?

Specific comments and technical corrections

- P2, line 21: consider changing "The advantage of the MAXDOAS technique in comparison to the traditional DOAS approach is that vertical column densities of several gases can be retrieved with some information on the vertical distribution" to "The advantage of the MAXDOAS technique in comparison to the zenith-sky DOAS approach is that vertical column densities can be retrieved with some information on the vertical distribution in the lower troposphere" - P2, line 29: "..satellite product and that of a chemical transport model" -> "and from results of a chemical..." - P2, line 29: " Tirpitz et al. (2020) found very good agreement" -> please quantify - P3, line 3: "The authors indicate that HCHO emitted by primary sources dominates .... HCHO decreases by approximately 1/3 in the afternoon": this is not what is seen in this study (Figure 3 shows larger HCHO in the afternoon). Could you comment this while presenting Figure 3? - P3, line 26 and P4, line 3: "records spectra at 0.075 cm−1 resolution" and "typically at 0.005 cm−1 resolution". What is the difference in resolution between the 2 FTIR instruments implying for HCHO measurements? Should we expect a difference in noise? Sensitivity? - P4, line 23: "azimuth angle of 85 with respect to the north": this means 85° E? but in line 27, the sequence starts first in the West and then to the East – please clarify (and add the azimuth measurements directions in Fig.1) - P4, line 27: how long is this measurement sequence taking in term of time? - P5, tabl1: give details of polynomial and offset - P5, line 9: why aerosols uses a Tikhonov regularization while trace gases retrievals uses optimal estimation? - P5, line 15: 338nm is not the middle of the 324.5-359nm interval (but it is close to it: 341nm!). What is the interval for the O4 SCD retrieval? - P5, end of Sect 2.2: give the HCHO MAXDOAS error estimations, as done at the end of Sect 3.1 for FTIR. - P6, figure 1: the numbers of the colorbar are difficult to read. - P6, line 17: "report values in the same order of magnitude, however, higher values in MAX-DOAS measurements than the FTIR instrument are apparent" -> from Figure 2 and 3, MAXDOAS data seems often about twice as large than the FTIR.... How would v1 and or v2 compare here? - P7, fig2: there seems to be much more variability in the MAXDOAS after 2015 compared to 2013 and 2014. Is there a reason for it? - P7, line 14: "nevertheless, the values do not differ significantly and

present similar seasonal cycles" – I would rephrase the "do not differ significantly" to something like "the 2 datasets are within each other error bars/temporal variability" - P8: are figures 3 and 4 only made with coincident hourly averages/months or with all the available datasets ? can this explain part of the variability? - P9, line 9: "measurements conducted towards both sides of the scanning plane." -> explain more how v3 data are retrieved. This is not so usual. - P9, line 30: "linear regression not constrained to zero is shown in red" in Figure 7, while it is green for the not constrained to zero for Figure 6. It is a bit perturbing. Keep same color conventions. - P11, line 2: give some numbers for the different DOF for v1, v2, v3. The differences (37% for v1 vs v3 and 28 % for v2 vs v3) seems a lot for a not so large difference in DOF seen in Figure 8, but numbers would help. Why DOF for v1 are so "not symmetrical" around 1? - P12, line 6: "how the retrievd profile" -> "how the retrieved profile" - P12, eq 2): explain bold vs non-bold "Xapr" - P13, line 1: "that AKtot is without units and shown" → as shown?! - P13, line 6: systematic -> systematic - P14, line 5: "After 12 h LT, conditions change so that larger HCHO VCDs are measured towards the western side of the scanning plane, peaking at 13-14 h." – can you put this in relation to distribution shown by OMI in Figure 1 (overpass around 13h30LT)"? - P14, figure 9: do you have information on wind conditions, to try to also separate/estimate possible contribution of different wind direction to the east-west difference during the day? - P.14, sect 3.3.3: try to compare FTIR to v1 in the morning and to v2 in the afternoon, when the sun is in the same direction that the MAXDOAS pointing direction. - P15, line 17: "Neither the retrieved FTIR profile nor the MAX-DOAS profile retrieval have sufficent degrees of freedom, therefore the strategy of using the profile information from one instrument together with the averaging kernel of the other instrument is not too promising." – reformulate. "sufficient degrees of freedom" to do what? Give values for the DOF! - P15, line 28: "So the average of $\Delta$colDOAS and $\Delta$colFTIR are zero." → the errors are not mentioned in the above paragraph, and these do not simplify one another, no? - P15, eq 8: end of the equation is missing: ". . ." - P16, figure 10: there seems to be a specific behavior for scatter plots at 12h and 13hLT, with a second "blob" of points not at all on the 1:1 line.

Can you comment this? - P17, line 6: "limited to just this hour" -> "limited to just one hour"? - P17, line 11 and 12: to my feeling, this sentence would be better suited after "The slope is given by the averaging kernels of the two instruments and the shape of the variable profile v. In Mexico City, we could assume that at 9 h LT the mixing layer is well mixed with HCHO up to a certain height with a constant concentration but with 0 or at least a constant HCHO value above this height. For this simple assumption (the only Eigenvector is constant in the mixing layer but 0 above it), the slope is the . . ." - P17, line 23: "therefore given due to the fact that" -> "therefore given by the fact that" - P17, sect 3.4: comment a bit more this background FTIR dataset (at least mentioning how it compares in Vigouroux et al., 2018 and 2020).

Suggested References:

Franco, B., Hendrick, F., Van Roozendael, M., Müller, J.-F., Stavrakou, T., Marais, E. A., Bovy, B., Bader, W., Fayt, C., Hermans, C., Lejeune, B., Pinardi, G., Servais, C., and Mahieu, E.: Retrievals of formaldehyde from ground-based FTIR and MAX-DOAS observations at the Jungfraujoch station and comparisons with GEOS-Chem and IMAGES model simulations, Atmos. Meas. Tech., 8, 1733–1756, https://doi.org/10.5194/amt-8-1733-2015, 2015.

---

## Author Comment (AC1) · 24 Nov 2020

**General comments**

Beyond Sect. 3.3.1, Sect. 3.3.2 and Sec. 3.3.3, one simplistic view of investigating the gathered data would be that, since FTIR performs direct Sun observations and DOAS V1 sense air masses towards the East, and DOAS V2 towards the West, the FTIR observations could be split between morning and afternoon observations in order to compare to V1 and V2 DOAS data, respectively.
For instance, once smoothed by the averaging kernel, have the authors compared the FTIR morning data to morning V1 DOAS, and FTIR afternoon data to afternoon V2 DOAS?
Would that compare better (or worse) to the FTIR am/pm data than V3?
**ANSWER:** Following both Referees suggestions, the comparison between FTIR and DOAS V1 during the morning and FTIR and DOAS V2 during the afternoon was conducted. The results, presented in Figure 5 (fourth row) reveal a better agreement between the two measurement techniques. The text of the caption of Figure 5 as well as the text of section 3.3.1 was modified accordingly.

The new caption reads as follows:

"Figure 5. Comparison between FTIR and MAX-DOAS measurements conducted at the UNAM measurement site. The first and second row panels correspond to the VCDs retrieved from MAX-DOAS measurements conducted towards the eastern (V1) and western (V2) measurement sides, respectively. For the third row panel, corresponding to the V3 data product, the VCDs are retrieved including both measurement sides. The fourth row panel corresponds to the comparison between FTIR and MAX-DOAS V1 during the morning and FTIR and MAX-DOAS V2 during the afternoon. The linear regression when forced to zero (red) and not constrained (green) is presented. Black lines represent the 1:1 relation. The left column shows the direct correlation between coincident pairs, whereas the middle column compares the retrieved FTIR VCDs with those calculated from the smoothed MAX-DOAS profiles using the averaging kernel from the FTIR (see text). The right column shows the total column averaging kernel of the FTIR (red lines) and MAX-DOAS (blue lines) retrievals. The dashed black lines on the first, second and fourth row represent the Averaging Kernel of V3."

The relevant text that was modified in section 3.3.1 can be identified in color green.

[Figure]

Modified Figure 5

Also, although it is mentioned throughout the manuscript, it is relevant that the authors state "up front" that, due to the different sensitivities of the techniques, the FTIR and the MAXDOAS information do not refer exactly to the same altitudes of the atmosphere (e.g., averaging kernel in Fig. 5).
**ANSWER:** This fact was stated in the abstract.

More than comparing one data set to the other, it might be more useful to use (and present) both data sets as complementary to each other. Also, the inhomogeneity of HCHO in Mexico City could be investigated even further by using the lowest elevation angles of the MAXDOAS data (i.e., near-surface HCHO) at the different azimuth angles (although that might result in another paper by itself).
**ANSWER:** We thank the Referee for this interesting suggestion, it was added in the Discussion and Conclusions section, as future work.

Also, in addition to the urban data, this work presents HCHO observations form the high-altitude site of Altzomoni (FTIR). Given the sparse number of measurements of HCHO performed from high-altitude locations, the manuscript would probably benefit if the authors could dig a bit further on these data set since, in fact, these data are merely presented in one small paragraph in the manuscript (Sec. 3.4.), but not really discussed (e.g., how do the HCHO VCD in clean and in urban scenario compare or the reasons behind the daily and seasonal evolution shown in Fig. 11), or put in context (even if briefly) with HCHO observations from other high altitude sites worldwide.
**ANSWER:** As suggested by both Referees, the dataset of Altzomoni was discussed in more detail in the manuscript and put in context with other high altitude sites worldwide and previous work.

The following text was added: "HCHO VCDs measured at Altzomoni are in the same order of magnitude as HCHO VCDs reported by Vigouroux et al. (2018) for several "clean" sites stations belonging to the NDACC network, such as Zugspitze, other mountain site (however at a latitude of 47° and an altitude of 3 km) as well as for Mauna Loa, at a latitude of 20° and an altitude of 3.4 km." … "Vigouroux et al. (2018) report the same behavior (a maximum in the late afternoon between 16 and 18 local time) for other stations of the NDACC network: Bremen, Paris, Toronto and Lauder. Further analysis should be conducted regarding the diurnal HCHO cycle at Altzomoni, however the detected maximum at late afternoon could be attributed to upslope transport or to secondary HCHO production that has reached a maximum at a certain hour of the day."

**Specific Comments**

P1, L11-12: "A time-dependent comparison revealed that the vertical distribution of this pollutant, guided by the evolution of the mixing layer height, can play an important role in how the results are affected."
Do the authors mean that the vertical distribution of HCHO can play an important role in how the results are affected? i.e., are the results affected by the distribution of HCHO?
Please, clarify.
**ANSWER:** The text was modified with the objective to clarify the idea. The following text replaced the original one: "The temporal change in the vertical distribution of this pollutant, guided by the evolution of the mixing layer height, affects the comparison of the two retrievals with different sensitivities (total column averaging kernels)."

P1, L21: HCHO is mainly (not "also") formed from the oxidation of CH4 and NMVOCs. This is particularly relevant for the Altzomoni data presented.
**ANSWER:** The change was made.

P4, L27: How long does it take to perform 1 scan (i.e., 90, 0, 2, 6, 13, 23, 36, 50, 65,82)?
**ANSWER:** With this setup, a complete scan takes about 7 min. This text was added to the manuscript.

P4, L31: In addition to the mentioned filters, is any sort of cloud filter applied to the DOAS data?
**ANSWER:** At the moment there is no cloud filter applied to the DOAS data, however, in this comparison exercise and since the DOAS data was compared to FTIR data, a cloud filter was inherently applied since the FTIR data was measured under no cloudy conditions. The application of a cloud filter to the DOAS data is work in progress.

Please, similarly to the error estimation offered for FTIR observations (P4, L 14), provide an error estimation of the MAXDOAS data presented in the manuscript.
**ANSWER:** The following text was added in section 2.2: "For the retrieved HCHO MAX-DOAS VCDs, the calculated noise error of the mean column is 5.8% while the systematic error due to uncertainty in the spectroscopy is 2.2%."

P5, L2: 324.6-359 nm is chosen for the HCHO spectral DOAS analysis. Why that particular range instead of the one suggested by e.g. Pinardi et al (2013)? Given the chosen spectral range, the possible spectral interference of BrO and/or O3 should be addressed (maybe a test with chosen days?) not only for the city data but also for the high-altitude observations. Also, what polynomial did the authors use for the HCHO DOAS retrieval? Note the impact of the polynomial mentioned by Pinardi et al. (2013).
**ANSWER:** The fitting settings used were following recommendations sent by Gaia Pinardi in November 2017 in the framework of the NIDFORVal project (S5P NItrogen Dioxide and FORmaldehyde VALidation). BrO and $O_3$ cross sections are included in the fitting. Two references, that refer to these settings, were added: Hendrick et al. 2016 and Pinardi 2017 (personal communication). A polynomial order 5 was used along with an offset order 1 (linear offset) (Hendrick et al., 2016; Pinardi, 2017). This text was added to the manuscript -in green since both Referees suggested this information should be added-.

P6, Sec. 3.1: Since Section 2 addressed FTIR and MAXDOAS observations, and Section 3 is entitled as "Results", probably Sect. 3.1 would make more sense after presenting the results of FTIR and MAXDOAS observations (Sect. 3.2).
**ANSWER:** As suggested by the reviewer, this section was placed after presenting the results of FTIR and MAX-DOAS observations.

P6, L14: why is V3 chosen (and not V1 and/or V2)? Since V1 and V2 are referred throughout the manuscript, the authors may want to include the time series of not only the V3 VCD in a figure (Fig. 2), but also of V1 and V2. Also, how is V3 retrieved? i.e., do the authors averaged the dSCDs observed at V1 and V2, and then invert V3 VCD from that averaged V1+V2 dSCD?
**ANSWER:** V3 was chosen because it has more information content than V1 and V2. The time series of V1 and V2 were included in the figure, as suggested by the reviewer. The MAX-DOAS dataset was updated to include data up to May 2020. The text in the manuscript and caption of Figure 2 (now Figure 1 due to the order change suggested by the reviewer in the previous point) was updated accordingly.

The following text was added in the manuscript, at the end of section 2.2 and just before section 3:

"Three different versions of HCHO VCDs were retrieved using the MMF code: **V1** retrieved VCDs from MAX-DOAS measurements conducted towards the east (telescope's azimuth angle of 85° with respect to the north), **V2** retrieved VCDs from MAX-DOAS measurements conducted towards the west (telescope's azimuth angle of 265° with respect to the north) and **V3** retrieved VCDs from MAX-DOAS measurements conducted towards both sides of the scanning plane. To simplify terminology, for the remainder of the manuscript version **V1** will be referred as "east", version **V2** will be referred as "west" and version **V3** will be referred as "both".

For **V1**, **V2** and **V3** the same *a priori* is used both for the trace gas and for the aerosol. For **V3**, the "scan" is simply treated as consisting of two different azimuth directions. The **V1**, **V2** and **V3** retrievals are performed independent of each other and differ in the definition of a "scan", where **V3** contains all pointing directions from **V1** and **V2** together. A single vertical profile is retrieved in both directions for **V3**, so assuming horizontal homogeneity. This assumption clearly is not fulfilled, however, it is also not fulfilled in a single viewing direction since the effective light path is around 5-20 km. As pointed out in the manuscript, the advantage of using both directions is a higher information content, the disadvantage is a more rigorous break down of the homogeneity assumption."

P7, figure 2: Based on the averaging kernels shown later in Fig. 5, it would be helpful to remind the reader the altitude ranges covered by each instrument (e.g., FTIR UNAM VCD 2-16 km; FTIR UNAM VCD 4-16 km; MAXDOAS UNAM VCD 2-5 km)
**ANSWER:** The following text was added to the caption of Figure 1 now (before Figure 2) due to the change order suggested by the Reviewer: "The altitude ranges covered by each instrument are FTIR UNAM VCDs 2-16 km, FTIR Altzomoni VCDs 4-16 km and MAX-DOAS UNAM VCDs 2-5 km."

P7, L4: "...and thus probes cleaner atmospheric columns" as long as there is no upslope transport (is there at Altzomoni?). Also, how do these VCD at Altzomoni compare to HCHO observations at other high altitude research sites? As mentioned in the general comments, authors are kindly advised to address further the results of Altzomoni data throughout the manuscript since those data are relevant by themselves (note the very scarce HCHO observations from high-altitude sites).
**ANSWER:** A maximum observed later in the afternoon at Altzomoni could be an indication of upslope. The results were further addressed  throughout the manuscript, especially in section 3.4. The text "as long as there is no upslope transport" was added.

P7, L6: "...in general larger than...". How much larger? Please, quantify.
**ANSWER:** The quantities were provided, the following text was added: "0 to 38% for V1, 15 to 47% for V2 and 29 to 61% for V3".

P7, L13: "...VCDs are larger than...". How much larger? Please, quantify.
**ANSWER:** The quantities were provided, the following text was added: "2 to 35% for V1, 17 to 51% for V2 and 23 to 75% for V3".

P9, Sect. 3.1.1: See general comments (i.e., are the FTIR morning data comparable to morning V1 DOAS, and FTIR afternoon data comparable to afternoon V2 DOAS?)
**ANSWER:** Following both Referees suggestions, the comparison between FTIR and DOAS V1 during the morning and FTIR and DOAS V2 during the afternoon was conducted. The results, presented in Figure 5 (fourth line) reveal a better agreement between the two measurement techniques. The text of the caption of Figure 5 as well as the text of section 3.3.1 was modified accordingly.

P9, L24: Please, specify (i.e., quantity) the (average) degrees of freedom (DOF) of the retrieved VCD for each technique (not only MAXDOAS but also FTIR). As for the DOAS V1, V2, V3 measurements,

do they have similar degrees of freedom? Figure 8 shows they are not the same, please provide an average DOF for V1, for V2 and for V3 or the time series.
**ANSWER:** The average degrees of freedom for V1, V2 and V3 as well as for both FTIR sites were added, the text was modified as follows "… less than two degrees of freedom (average values being 0.692 for V1, 0.782 for V2 and 0.970 for V3) and do not represent the true atmospheric profile, while the average FTIR degrees of freedom is 1.0 for the UNAM site and 1.1 for the Altzomoni site."

P10, Fig. 5: how is the vertical grid of the FTIR inversion compared to the one used for the MAXDOAS inversion?
**ANSWER**: This information was added to Figure 6, as a complement (Figure 6b) of the figure already presented.

P13, L8-9: "the retrieved profile using both sides of the measurement plane is to our current knowledge the best estimation for the HCHO profile" if one assumes horizontal homogeneity (?).
**ANSWER:** as suggested by the Referee, the phrase "(if one assumes horizontal homogeneity)" was added after the word profile.
We would like to refer to the Referee to the paragraphs in color green where we provide an answer to comment "P6, L14", where we give estimates of the effective distance and point out that horizontal homogeneity is already not fulfilled for V1 and V2.

P13, L19: please provide a DOAS dSCD error estimation.
**ANSWER:** For the lower elevation angles (where the HCHO signal is higher), the DOAS dSCD error is typically 15%.

P15, L21: "...both instruments are measuring coincidently the same atmospheric state" Would that be true given Fig. 1? Probably only if V3 is used as measuring vector (?)
**ANSWER:** The formulation "taking into account" was replaced by "assuming".

**Technical corrections**

P1, L 3-6: For the MAX-DOAS measurements, the software QDOAS was used to calculate differential Slant Column Densities (dSCDs) from the measured spectra and subsequently the Mexican MAX-DOAS Fit retrieval code (MMF) to convert from dSCDs to Vertical Column Densities (VCDs). The direct-solar absorption spectra measured with FTIR were analyzed using the PROFFIT retrieval code These sort of details would be better if included later in the text (Sec. 2.1, 2.2), not in the abstract.
**ANSWER:** These details were included in the methodology as well. In order to provide as many details as possible, we thought it would be a good idea to also include them in the abstract.

P1, L 10: "could demonstrate"
**ANSWER:** The change was made.

Probably more accurate would be "suggests" or "indicates"
**ANSWER:** The change was made.

P1, L 12: "Apart from the reported..."
In addition to the reported...
**ANSWER:** The change was made.

P2, L 12: "... satellites, aircraft, vehicles or ground based"
Balloons as well
**ANSWER:** Balloons was added to the text.

P2, L 20: Since CINDI is mentioned, probably the authors should also mention the more CINDI2 campaign (Kreher et al., 2020)
**ANSWER:** As suggested by the Referee, the CINDI-2 campaign was included as well.

P2, L26: LP in LP-DOAS was not introduced before (i.e., long-path DOAS)
**ANSWER:** Long Path was defined, as suggested by the Referee.

P2, L24-28: FTIR vs MAXDOAs literature. The authors may want to consider including Franco et al., AMT, 2015;
**ANSWER:** The Franco et al. (2015) reference was added, as suggested by both Reviewers.

P2, L30: "Another study by Garcia et al. ....dominate the HCHO concentration at the surface". The authors may want to split that long sentence.
**ANSWER:** The long sentence was splitted as suggested by the Referee.

P3, L5: In which sense is the work presented an "unprecedented comparison"? Is it due to the length of the study (i.e., 6 years)? Is it due to the location of the study (i.e., Mexico)?
**ANSWER:** We believe both, the following text was added after unprecedented comparison "(in terms of length and location)"

P3, L7: "to characterize the difference" in?
**ANSWER:** "in both measurement techniques" was added.

P3, L5-12: To ease the reader, the authors may want to specify in which section will be addressed each of the topics mentioned in this paragraph.
**ANSWER:** As suggested by the Referee, the sections where each of the specific topics mentioned in the paragraph were addressed, were added to the text.

P3, L15: "One" of the sites "is..."
**ANSWER:** The change was made, as suggested by the Reviewer.

P3, L17: "The other" site "is the Altzomoni..."
**ANSWER:** The change was made, as suggested by the Reviewer.

P3, L21: At Altzomoni, please specify that the FTIR is part of NDACC. Note that NDACC also comprises DOAS instruments.
**ANSWER:** The clarification was made in the text.

P5, L4: Even if it is mentioned by Friedrich et al., to ease the reader the authors may want to include at least the spectral range where O4 is retrieved.
**ANSWER:** The spectral range where $O_4$ is retrieved was included, the following text was added "in the 336 to 390 nm wavelength range,"

P6, Fig. 1: Please, enlarge the size letter of the two sites in the map, they are hard to see. Also, a circle indicating the region that comprises MCMA would be helpful for the reader.
**ANSWER:** The size letter of the two sites in the map was enlarged and the region comprising the MCMA was indicated (in white).

P7, L11: "the seasonal HCHO VCDs" Are those the monthly average data?

**ANSWER:** Yes, they are monthly average data. The clarification was made in the text.

P7, L12: The meaning of the black line is not clear. Also, does it relate to FTIR or to MAXDOAS? Note that in Fig. 4 the black data are very hard to distinguish.
**ANSWER:** It refers to both FTIR and MAX-DOAS. The black values were plotted to show the average of month and standard error based on monthly means. For the sake of clarity, and since the results of MAX-DOAS V1 and MAX-DOAS V2 were added to the figure, as suggested by Referee 2, the black lines were removed from the figure.

P13, L13: What do the authors mean with "the calculation of the red trace"? Do they mean "line"? Note that the equations in page 13 refer to matrices and the word "trace" might be misleading. If refer to line, authors are advised also to change it in the caption of figure 9.
**ANSWER:** Yes, trace means line. In all the document the word trace was changed for line.

P14, Fig. 9: A horizontal line at VCD difference = 0 might help the reader to understand that figure.
**ANSWER:** A horizontal line was added to VCD difference = 0

P15, L25: "i" stands for?
**ANSWER:** "*i*" is the index for the hours for which there exist coincident measurements. This information was added to the text.

P15, Eq 8: Equation incomplete:
**ANSWER:** It was most probably not appropriate using the "...", we have removed them and write explicitly in the line before what we assume:
"… So the average of $\Delta colDOAS$ and $\Delta colFTIR$ are zero. In addition we assume that the errors epsilon_FTIR(i), epsilon_DOAS(i) are independent and in average zero, we assume also that they are independent with respect to AK_DOAS(i), AK_FTIR(i), Xtrue(i), so that we can simplify the calculation of equation (7) to equation (8)."

P17, L15-18: "The slope is given by the averaging kernels of the two instruments and the shape of the variable profile v, and for the simple assumption described above, that the only Eigenvector is constant in the mixing layer but 0 above it, the slope is the fraction of the mean averaging kernel elements in the mixing layer (MAXDOAS/FTIR)." The authors may want to split this very long sentence.
**ANSWER:** This very long sentence was split.

P17, L21: Given all the assumptions needed, more than "to demonstrate", probably it would be better "to support"
**ANSWER:** The change was made, according to the Referee suggestion.

P17, L21-24: The paragraph is a bit confusing. The authors may want to clarify what they mean.
**ANSWER:** The paragraph was reformulated and hopefully it is now more comprehensible: "The individual plots in Figure 10, showing the correlations and their slopes for each hour, allow us to support the fact that instead of simply cross validating the FTIR and MAX-DOAS retrievals, it is possible to assume that the mixing layer height dominates the variability and that such simplification is valid on a certain hour. The validation is therefore given by the fact that a plausible variability for each hour explains the slope and correlation for different hours, rather than that the slope and the correlation is close to 1.0."

P18, L20: "...depending on atmospheric conditions ..." and the wavelength.
**ANSWER:** The change was made, according to the Referee suggestion.

P19, L28-36: This paragraph presenting the megacity of Mexico might fit better in the introduction (Sect. 1).
**ANSWER:** The paragraph was moved to the introduction, as suggested by the Referee.

---

## Author Comment (AC2) · 25 Nov 2020

**General comments**

The datasets are important (in length and for their high HCHO VCD columns) and the comparison of MAXDOAS and FTIR is of great interest, and they would deserve some more explanations. FTIR datasets have already been used in other publications (e.g., Vigouroux et al., 2018; 2020) and this should be emphasized a bit more, putting the 2 sites in the context of other existing HCHO FTIR. Also the Altozomoni site is showed in Sect. 3.4, but only very shortly.
**ANSWER:** Following the Referee suggestion the use of FTIR datasets used in other publications was emphasized and discussion was added to results presented from the Altzomoni site.

The following text was added: "HCHO VCDs measured at Altzomoni are in the same order of magnitude as HCHO VCDs reported by Vigouroux et al. (2018) for several "clean" sites stations belonging to the NDACC network, such as Zugspitze, other mountain site (however at a latitude of 47° and an altitude of 3 km) as well as for Mauna Loa, at a latitude of 20° and an altitude of 3.4 km." … "Vigouroux et al. (2018) report the same behavior (a maximum in the late afternoon between 16 and 18 local time) for other stations of the NDACC network: Bremen, Paris, Toronto and Lauder. Further analysis should be conducted regarding the diurnal HCHO cycle at Altzomoni, however the detected maximum at late afternoon could be attributed to upslope transport or to secondary HCHO production that has reached a maximum at a certain hour of the day."

MAXDOAS HCHO data from Mexico is presented here for the first time (to my knowledge), and these datasets (v1, v2 and especially v3) need a bit more explanations. In Sect. 2.2, the MAXDOAS error estimation are not even mentioned! Information on the polynomial and offset choice in Table 1 are missing, and more explanations of the v3 analysis should be given. It is said that "VCDs retrieved using measurements from both sides of the scanning plane are in general larger than VCDs retrieved using data from measurements of only one of the sides. This result can be explained by the larger amount of information available for the retrievals when dSCDs in different elevation angles and both scanning directions are used" (end of P.9 and P. 11), but it is never mentioned how this v3 is done. How are the opposite directions treated in term of apriori, aerosols content,...? Is the retrieval considering an homogeneous atmosphere for the retrieval, or are the differences for v1 and v2 somehow taken into account for the v3 retrieval? An illustration of the behaviors of v1, v2 and v3 on a typical day would be a nice addition (and also adding v1 and v2 on the diurnal and seasonal figures 3 and 4). Also the degrees of freedom should be quantified (numbers in figure 8 are small and difficult to read).
**ANSWER:** To address these issues, we now provide the MAX-DOAS error estimation as well as details regarding the polynomial and offset used for the DOAS HCHO retrieval. More explanation of the V1, V2 and V3 analysis is now given. The degrees of freedom for MAX-DOAS and for FTIR are now provided and the font size of Figure 8 was enlarged.

Regarding the specific explanation the following text was added in the manuscript, at the end of section 2.2 and just before section 3:

"Three different versions of HCHO VCDs were retrieved using the MMF code: **V1** retrieved VCDs from MAX-DOAS measurements conducted towards the east (telescope's azimuth angle of 85° with respect to the north), **V2** retrieved VCDs from MAX-DOAS measurements conducted towards the west (telescope's azimuth angle of 265° with respect to the north) and **V3** retrieved VCDs from MAX-DOAS measurements conducted towards both sides of the scanning plane. To simplify terminology, for the remainder of the manuscript version **V1** will be referred as "east", version **V2** will be referred as "west" and version **V3** will be referred as "both".

For **V1**, **V2** and **V3** the same *a priori* is used both for the trace gas and for the aerosol. For **V3**, the "scan" is simply treated as consisting of two different azimuth directions. The **V1**, **V2** and **V3** retrievals are performed independent of each other and differ in the definition of a "scan", where **V3** contains all pointing directions from **V1** and **V2** together. A single vertical profile is retrieved in both directions for **V3**, so assuming horizontal homogeneity. This assumption clearly is not fulfilled, however, it is also not fulfilled in a single viewing direction since the effective light path is around 5-20 km. As pointed out in the manuscript, the advantage of using both directions is a higher information content, the disadvantage is a more rigorous break down of the homogeneity assumption."

As suggested by the Referee, V1 and V2 were added to the diurnal and seasonal cycles figures (now Figure 2 and Figure 3 due to a reorganization of figures suggested by Referee 1).

Add reference and discussion of Vigouroux et al., 2009 (Reunion Island), and Franco et al., 2015 (Jungfraujoch) to better highlight the difference in sensitivities between MAXDOAS and FTIR. The AVK are shown in the last column of figure 5, but they maybe deserve a specific figure instead, comparing the AVK of FTIR, MAXDOAS v1, v2 and v3 on the same panel. When seeing the AVK, the 2 measurements are not sensitive at all at the same altitudes, so if the HCHO is not homogeneously distributed, we don't expect the same measurements.
**ANSWER:** The Vigouroux et al. (2009) reference is already in the text. The Franco et al. (2015) reference was added to the manuscript, as suggested by both Referees.
For the sake of clarity, we suggest that showing the AVKs in Figure 5 provide complementary information while analyzing the different comparisons between FTIR and MAX-DOAS and for this reason we have decided to leave the AVKs panels in Figure 5, but we added a dashed line of the AVK of the version 3 to all AKV panels. An error was found in the program plotting the total column AVKs of MAX-DOAS, this was corrected and is now presented in the updated figure. More explanation was given to the differences of AVKs for FTIR and MAX-DOAS V1, V2, V3 and the new comparison: V1 vs FTIR during the morning and V2 vs FTIR during the afternoon. The modified Figure 5 is shown below:

[Figure]

Modified Figure 5

The comparisons between MAXDOAS and FTIR are a bit perturbing, as Figures 2 to 4 seems to suggest a bias of 50%, when all the data are considered (are the MAX-DOAS data cloud filtered?), while figure 5, when the coincident measurements are selected, seems to indicate smaller bias (28% if considering v3 with the slope passing by the origin – although the non-forced regression show an important systematic intercept). The different regressions of Fig. 5, should be discussed in more details. Also figure 2 shows better MAX-DOAS to FTIR comparison during 2013-2014, while since 2015 the MAX-DOAS are up to twice the FTIR values. Is there a reason for that? Has the instrument or the measurement strategy changed?

**ANSWER:** Figure 2 (now Figure, 1 due to a change order suggested by Referee 1) includes all measurements. In this first figure, data is not filtered as everything that is available for both measurement techniques is presented. The different regressions on Figure 5 were discussed in more detail.

The MAX-DOAS measurement strategy was changed in 2015, adding more measurements at lower elevation angles. We would like to point out that adding more measurements at low elevation angles enhances the sensitivity at lower altitudes, where the FTIR has lower sensitivity and hence, capturing better concentrations of HCHO at low altitudes with MAX-DOAS will result in an increase in the bias.

Instead of (or in addition to) comparing hourly MAXDOAS v3 to FTIR in Figure 10, why not compare the morning FTIR data with the MAXDOAS measuring to the East, and afternoon FTIR to MAXDOAS measuring to the West? Adding the measurement directions to the map of Figure 1 could help the reader understanding the measurements at each site. Giving some explanations on the inhomogeneities in HCHO seen by OMI could also help the reader (is there any specific vegetation? Industry? How is the orography around Mexico city? Can some HCHO be "trapped" by winds and terrain?). Are the conclusion of Section 3.3.2, with the larger abundances on the eastern side of the scanning plane during the morning hours, and a change after 12h LT, supported e.g. by wind direction changes? How is this gradient explained? Are the other MAX-DOAS Mexican sites HCHO measurements confirming this horizontal inhomogeneities?

**ANSWER:** Following both Referees suggestions, the comparison between FTIR and DOAS V1 during the morning and FTIR and DOAS V2 during the afternoon was conducted. The results, presented in Figure 5 (fourth line) reveal a better agreement between the two measurement techniques. The text of the caption of Figure 5 as well as the text of section 3.3.1 was modified accordingly.

The measurement directions were added on the map of Figure 1 (now Figure 2 due to the re-ordering suggestion from Referee 1). While discussing Figure 9, information was added regarding the orography around Mexico City and how it can impact wind patterns and transport in the basin.

Also, a more fundamental question. There has been recent studies (e.g. with Pandora, https://amt.copernicus.org/preprints/amt-2020-158/) showing contamination of "plastic" material from the instrument, emitting HCHO in case of hot temperatures – is this eventuality been excluded here? Is there any relation of the measured HCHO with the temperature?

**ANSWER:** The interference by HCHO in the PANDORA instrument was, according to our knowledge, due to the use of the material Delrin, we are using Nylamid. In the case of using any thermoplastic (as both PANDORA and MAX-DOAS are using), the possibility to have interference of volatile organic compounds is all the time possible. In the case of the MAX-DOAS measurements, and due to the fact that the reference ("zenith measurement") is taken only a few minutes before the elevation angles measurements, we consider that this effect, if exists, is not an issue, since it can be considered to be canceled out because all the measurements conducted during a same cycle (that lasts only a few minutes) would have the same influence.

**Specific comments and technical corrections**

- P2, line 21: consider changing "The advantage of the MAXDOAS technique in comparison to the traditional DOAS approach is that vertical column densities of several gases can be retrieved with some information on the vertical distribution" to "The advantage of the MAXDOAS technique in comparison to the zenith-sky DOAS approach is that vertical column densities can be retrieved with some information on the vertical distribution in the lower troposphere"
**ANSWER:** The change was made, as suggested by the Referee.

- P2, line 29: "..satellite product and that of a chemical transport model" -> "and from results of a chemical..."
**ANSWER:** The change was made, as suggested by the Referee.

- P2, line 29: " Tirpitz et al. (2020) found very good agreement" -> please quantify
**ANSWER**: The following text was added "(an average root-mean-square difference of $1.4 \times 10^{15}$ molec/cm$^2$ )".

- P3, line 3: "The authors indicate that HCHO emitted by primary sources dominates .... HCHO decreases by approximately 1/3 in the afternoon": this is not what is seen in this study (Figure 3 shows larger HCHO in the afternoon). Could you comment this while presenting Figure 3?
**ANSWER:** We commented on this fact while presenting Figure 3 (now Figure 2). In general terms the observed differences in both studies: a decrease of HCHO in the afternoon reported by the Lei et al. (2009) study and the MAX-DOAS datasets reporting an increase of HCHO after 16 h in this study could be attributed to the duration of the experiment reported by Lei et al. (2009) which is a three-day modeled episode constrained by ground-based measurements conducted in 2003 and the length of the datasets that are being reported in the current study (more than 8 years of data between 2013 and 2020). In addition the measurement strategies are different, in this study we are reporting the amount of HCHO in the entire tropospheric column, while in the Lei et al. (2009) study the HCHO surface concentration (measured and modeled) is being reported.

- P3, line 26 and P4, line 3: "records spectra at 0.075 cm-1 resolution" and "typically at 0.005 cm-1 resolution". What is the difference in resolution between the 2 FTIR instruments implying for HCHO measurements? Should we expect a difference in noise? Sensitivity?
**ANSWER:** The difference is that for the higher resolution instrument the DOF improves by 0.1, as the instrument located at Altzomoni has a higher resolution (longer path length), but because the instrument is also more stable and better, errors might be smaller in Altzomoni as well. For more details please see Vigoroux et al., (2018). Blumenstock et al. 2020 have shown that the error due to channeling is a very important error source for a weak absorber as HCHO in NDACC-FTIR instruments as the one collocated in Altzomoni, such a detailed study is not available for the Vertex 80 instrument. The use of calculated hourly mean for the comparison reduces the random error in the measurements for the instrument located in Mexico City, where we measure almost ten times more spectra in this hour.

- P4, line 23: "azimuth angle of 85 with respect to the north": this means 85° E? but in line 27, the sequence starts first in the West and then to the East – please clarify (and add the azimuth measurements directions in Fig.1)
**ANSWER:** As suggested by the Referee the measurement directions were added to Figure 1 (now Figure 4 due to changes suggested by Referee 1) and a clarification regarding the measurement

directions was made. The text where this is clarified is presented at the end of section 2.2 (in color green).

- P4, line 27: how long is this measurement sequence taking in term of time?
**ANSWER:** With this setup, a complete scan takes about 7 min. This text was added to the manuscript.

- P5, tabl1: give details of polynomial and offset
**ANSWER:** A polynomial order 5 was used along with an offset order 1 (linear offset) (Hendrick et al., 2016; Pinardi, 2017). This information was added to the text before Table 1 -in green since both Referees suggested this information should be added-.

- P5, line 9: why aerosols uses a Tikhonov regularization while trace gases retrievals uses optimal estimation?
**ANSWER:** This is a very good question. We have a quite inhomogeneous mix of constraints and retrieval strategies, FTIR-retrievals (Vertex/UNAM and NDACC/Altzomoni) are done with Tikhonov constraint and the MAX-DOAS uses Tikhonov for the aerosol retrieval but OET optimal estimation for the gas retrieval.

First we would like to point out, that in this work we try to use the Averaging Kernel and *a prioris* to correct *a posteriori* for the different retrieval strategies, as the real information of the measurement should not depend too much on the used retrieval strategy, at least if the result is used carefully with the AVK and it's *a priori*.
Tikhonov constrains the form, while OET constrains typically the values.

As described in Friedrich et al. 2019, the *Sa* for $NO_2$ was constructed from a model simulation of gas concentration profiles in Mexico City. The concentration of the model could be compared to in situ measurements (at least near the ground) and it might be plausible to use it as *a priori*.
For gases there is *a priori* information available which is more related to the value than to their distribution.

We did not have simulation for aerosol concentrations, and the absolute value of the aerosol concentration in a certain altitude is not measured anywhere.

We have a good estimation of the diurnal behavior of the profile shape of the vertical aerosol distribution but not the values. Since using a Tikhonov regularization restricts the shape, as mentioned above, it seems more fitted to use this regularization for the aerosol retrieval instead of optimal estimation.

The *a priori* of the aerosol is *a priori* for each hour of the day. The hourly *a prioris* are constructed from long term ceilometer data and a measured climatology. The ceilometer measurement and the obtained climatology of the aerosol distribution is described by Garcia-Franco et al. 2018

For trace gases on the other hand, we do not have such a diurnal climatology based on measurements. Hence the optimal estimation method is better suited. Although we have model estimations for each hour, we choose to not use a time dependent a priori and instead include this variability via a better estimation of the SA matrix.

García-Franco, J.L., Stremme, W., Bezanilla, A. Ruiz-Angulo, A., Grutter, M. (2018) Variability of the Mixed-Layer Height Over Mexico City. Boundary-Layer Meteorology 167, 493–507. https://doi.org/10.1007/s10546-018-0334-x

- P5, line 15: 338nm is not the middle of the 324.5-359nm interval (but it is close to it: 341nm!). What is the interval for the O4 SCD retrieval?
**ANSWER:** The phrase was modified to "in between the range of the wavelength interval used for the QDOAS retrieval". The interval for the $O_4$ dSCDs retrieval is 336 to 390 nm, this information was added to the text as well.

- P5, end of Sect 2.2: give the HCHO MAXDOAS error estimations, as done at the end of Sect 3.1 for FTIR. -
**ANSWER:** The following text was added in section 2.2: "For the retrieved HCHO MAX-DOAS VCDs, the calculated noise error of the mean column is 5.8% while the systematic error due to uncertainty in the spectroscopy is 2.2%."

- P6, figure 1: the numbers of the colorbar are difficult to read.
**ANSWER:** The font size of the numbers of the colorbar was increased.

- P6, line 17: "report values in the same order of magnitude, however, higher values in MAX-DOAS measurements than the FTIR instrument are apparent"-> from Figure 2 and 3, MAXDOAS data seems often about twice as large than the FTIR.... How would v1 and or v2 compare here?
**ANSWER:** The differences between values were quantified and were added to the text, they are reported for V1, V2 and V3 using FTIR as the reference.

- P7, fig2: there seems to be much more variability in the MAXDOAS after 2015 compared to 2013 and 2014. Is there a reason for it?
**ANSWER:** In 2015 more elevation angles were added to the measurement sequence so that more measurements would be taken at lower elevation angles. We would like to point out that adding more measurements at low elevation angles enhances the sensitivity at lower altitudes, where the FTIR has lower sensitivity and hence, capturing better concentrations of HCHO at low altitudes with MAX-DOAS will result in an increase in the bias.

- P7, line 14: "nevertheless, the values do not differ significantly and present similar seasonal cycles" – I would rephrase the "do not differ significantly" to something like "the 2 datasets are within each other error bars/temporal variability" -
**ANSWER:** The change was made following the Referee recommendation.

P8: are figures 3 and 4 only made with coincident hourly averages/months or with all the available datasets ? can this explain part of the variability?
**ANSWER:** Figures 3 and 4 were made with all the available datasets and as the Referee suggests, part of the variability could be explained by this.

- P9, line 9: "measurements conducted towards both sides of the scanning plane." -> explain more how v3 data are retrieved. This is not so usual.

**ANSWER**: The explanation of the retrieval of V3 was added earlier in the manuscript, at the end of section 2.2.

- P9, line 30: "linear regression not constrained to zero is shown in red" in Figure 7, while it is green for the not constrained to zero for Figure 6. It is a bit perturbing. Keep same color conventions.
**ANSWER:** As suggested by the Referee, the color of the linear regression line was changed to green in order to keep the same color conventions..

- P11, line 2: give some numbers for the different DOF for v1, v2, v3. The differences (37% for v1 vs v3 and 28 % for v2 vs v3) seems a lot for a not so large difference in DOF seen in Figure 8, but numbers would help. Why DOF for v1 are so "not symmetrical" around 1?
**ANSWER:** The average degrees of freedom for V1, V2 and V3 were added, the text was modified as follows "… are used (average values being 0.692 for V1, 0.782 for V2 and 0.970 for V3) (Figure 8)..."
In addition Figure 8 was modified so that the font size of both the x and y axis was larger so that numbers could be more clearly observed.

Regarding the skewness of degree of freedom values for V1:
Asymmetry might be due to rejection of spectra is not symmetrically distributed, rejection occurs typically if there are obstacles or saturation, as the viewing angle is close to the sun angle.

- P12, line 6: "how the retrievd profile" -> "how the retrieved profile"
**ANSWER:** The correction was made, as suggested by the Referee.

- P12, eq 2): explain bold vs non-bold "Xapr"
**ANSWER:** All "Xapr" should be bold. The change was made.

- P13, line 1: "that AKtot is without units and shown" ! as shown?! -
**ANSWER:** We use the suggestion "as shown"

P13, line 6: systematic -> systematic
**ANSWER:** The correction was made, as suggested by the Referee.

- P14, line 5: "After 12 h LT, conditions change so that larger HCHO VCDs are measured towards the western side of the scanning plane, peaking at 13-14 h." – can you put this in relation to distribution shown by OMI in Figure 1 (overpass around 13h30LT)"?
**ANSWER**: As suggested by the Referee this statement was put in relation to the distribution shown by OMI, the following text was added: "The average HCHO distribution over the MCMA, reconstructed from OMI data (Figure 4), provides evidence of a larger enhancement of HCHO columns towards the western side of the MCMA at OMI overpass time, coinciding with our findings in terms of the identified horizontal inhomogeneity as well as timing."

- P14, figure 9: do you have information on wind conditions, to try to also separate/estimate possible contribution of different wind direction to the east-west difference during the day?
**ANSWER:** We have meteorological information at ground level, however since we are reporting the amount of HCHO in the entire atmospheric column, the representativeness of ground-based

meteorological information could be limited while trying to estimate or separate the possible contribution of different wind directions. The following text was added to the comments of Figure 9:

"The observed changes could also be related to orographic and meteorological conditions. Fast et al. (2007) report that surface wind measurements over the city indicate the production of strong convergence in the basin during the late afternoon, created by opposing propagating density currents and a gap flow originating in the southeastern corner of the basin. The authors conclude that in the MCMA short-range transport can be produced by the complex terrain surrounding it, producing local and regional circulations."

Fast, J. D., de Foy, B., Acevedo Rosas, F., Caetano, E., Carmichael, G., Emmons, L., McKenna, D., Mena, M., Skamarock, W., Tie, X., Coulter, R. L., Barnard, J. C., Wiedinmyer, C., and Madronich, S.: A meteorological overview of the MILAGRO field campaigns, Atmos. Chem. Phys., 7, 2233–2257, https://doi.org/10.5194/acp-7-2233-2007, 2007.

- P.14, sect 3.3.3: try to compare FTIR to v1 in the morning and to v2 in the afternoon, when the sun is in the same direction that the MAXDOAS pointing direction.
**ANSWER:** Following both Referees suggestions, the comparison between FTIR and DOAS V1 during the morning and FTIR and DOAS V2 during the afternoon was conducted. The results, presented in Figure 5 (fourth line) reveal a better agreement between the two measurement techniques. The text of the caption of Figure 5 as well as the text of section 3.3.1 was modified accordingly.

- P15, line 17: "Neither the retrieved FTIR profile nor the MAX-DOAS profile retrieval have sufficent degrees of freedom, therefore the strategy of using the profile information from one instrument together with the averaging kernel of the other instrument is not too promising." – reformulate. "sufficient degrees of freedom" to do what? Give values for the DOF!
**ANSWER:** Values for the DOF of the MAX-DOAS were given in section 3.3.1. In addition the text was clarified as follows: "Neither the retrieved FTIR profile (1.1) nor the MAX-DOAS profile retrieval (<2) have sufficient degrees of freedom to consider the retrieval as profile retrieval, therefore the strategy of using the profile information from one instrument together with the averaging kernel of the other instrument is not too promising."

- P15, line 28: "So the average of DcolDOAS and DcolFTIR are zero." ! the errors are not mentioned in the above paragraph, and these do not simplify one another, no?
**ANSWER:** In order to improve the explanation, the text before Equation 7 was modified as follows:
"The average of the product of the columns of both instruments is theoretically given by the following equation, for that purpose we introduce the errors epsilon_FTIR(i), and epsilon_DOAS(i), where *i* is the index which identifies a certain hour:"

And before Equation 8 we added the following text:
"… So the average of $\Delta$colDOAS and $\Delta$colFTIR are zero. In addition we assume that the errors epsilon_FTIR(i), epsilon_DOAS(i) are independent and in average zero, we assume also that they are independent with respect to AK_DOAS(i), AK_FTIR(i), Xtrue(i), so that we can simplify the calculation of equation (7) to equation (8)."

- P15, eq 8: end of the equation is missing: "..."
**ANSWER:** It was unlucky using the "...", we have removed them and write explicitly in the line before what we assume:

… So the average of ΔcolDOAS and ΔcolFTIR are zero. In addition we assume that the errors epsilon_FTIR(i), epsilon_DOAS(i) are independent and in average zero, we assume also that they are independent with respect to AK_DOAS(i), AK_FTIR(i), Xtrue(i), so that we can simplify the calculation of equation (7) to equation (8).

- P16, figure 10: there seems to be a specific behavior for scatter plots at 12h and 13hLT, with a second "blob" of points not at all on the 1:1 line. Can you comment this?
**ANSWER:** Good observation:
We produced the same plot for V1 and V2, but it seems not to be related to one special site.
around noon. So these values might be responsible for the larger variability in the V3 and are not easily to be explained by the sampling of different air masses.
The average sensitivity of version v3 near the ground is slightly larger than v2 and v1, which would explain a factor of up to 1.5 to pollution spikes near the ground, but not the observed factor of larger than 2.

- P17, line 6: "limited to just this hour" -> "limited to just one hour"?
**ANSWER:** The change was made, as suggested by the Referee.

- P17, line 11 and 12: to my feeling, this sentence would be better suited after "The slope is given by the averaging kernels of the two instruments and the shape of the variable profile v. In Mexico City, we could assume that at 9 h LT the mixing layer is well mixed with HCHO up to a certain height with a constant concentration but with 0 or at least a constant HCHO value above this height. For this simple assumption (the only Eigenvector is constant in the mixing layer but 0 above it), the slope is the ..." -
**ANSWER:** The change was made according to the Referee's suggestion.

P17, line 23: "therefore given due to the fact that" -> "therefore given by the fact that" -
**ANSWER:** The change was made, as suggested by the Referee.

P17, sect 3.4: comment a bit more this background FTIR dataset (at least mentioning how it compares in Vigouroux et al., 2018 and 2020).
**ANSWER:** As suggested by both Referees, the dataset of Altzomoni was discussed in more detail in the manuscript and put in context with other high altitude sites worldwide and previous work.

**Suggested References:**
Franco, B., Hendrick, F., Van Roozendael, M., Müller, J.-F., Stavrakou, T., Marais, E. A., Bovy, B., Bader, W., Fayt, C., Hermans, C., Lejeune, B., Pinardi, G., Servais, C., and Mahieu, E.: Retrievals of formaldehyde from ground-based FTIR and MAX-DOAS observations at the Jungfraujoch station and comparisons with GEOSChem and IMAGES model simulations, Atmos. Meas. Tech., 8, 1733–1756, https://doi.org/10.5194/amt-8-1733-2015, 2015.